



# An analytical solution for wind deficit decay behind a wind energy converter using momentum flux conservation validated by UAS data

Moritz Mauz[1], Bram van Kesteren[1], Andreas Platis[1], Stefan Emeis[2], and Jens Bange[1]

[1]Centre for Applied Geoscience, University of Tübingen, 72074 Tübingen, Germany
[2]Institute for Meteorology and Climate Research, Karlsruhe Institute of Technology, 82467 Garmisch-Partenkirchen, Germany

**Correspondence:** Moritz Mauz (moritz.mauz@uni-tuebingen.de)

**Abstract.** The wind deficit behind a wind energy converter (WEC) results from a complex interaction of forces. Kinetic energy is removed from the atmosphere, but coherent turbulent structures prevent a swift compensation of momentum within the wake behind the WEC. A detailed description of the wake is beneficial in meso-scale weather forecast (e.g. WRF models) and numerical simulations of wind wake deficits. Especially in the near to intermediate wake ($0.5 - 5$ rotor diameter D), the dominating processes characterising the wake formation change along the wake. The conservation equation of momentum is used as a starting point to map the most important processes assuming the WEC operates at maximum efficiency in a neutral stratified boundary layer. The wake is divided into three different regions to accommodate the changing impact of atmospheric turbulence and the shear created by the WEC onto the wake. A differential equation that depicts the variable momentum transport into the wind deficit along the wake is derived and solved analytically. Additionally, a numerical solution (Euler method) of the simplified momentum conservation equation is shown to provide a quality control and error estimate to the analytical model. The analytical solution is compared to in-situ wake measurements behind an Enercon E-112 converter, located in the Jade Wind Park near the North Sea coast in Germany, captured by the MASC-3 UAS (unmanned aircraft system) of the University of Tübingen. The obtained UAS data cover the distance from $0.5 - 5$ D at hub height behind the nacelle. The analytical and numerical model are found to be in good agreement with the data of the three measurement flights behind the WEC.

## 1 Introduction

Wind energy converters (WECs) remove kinetic energy from the mean flow in the atmosphere which results in a wind deficit in the wake behind each converter. The interaction of these individual wakes with the atmosphere and even with neighbouring converters needs to be understood, for example in order to increase wind park efficiency. With increasing sizes of WECs and wind farms the wind deficit created by the wind parks and individual converters affects the environment increasingly. The WIPAFF (Wind PArk Far Field) project determined the length of single wind park wakes and found their dimensions large





enough to influence local wind conditions (Platis et al., 2018; Bärfuss et al., 2019). Therefore, an influence on local weather (and neighbouring wind farms) is likely. WRF (weather research and forecasting model) simulations by Siedersleben et al. (2018) display the wind deficit using wind farm parametrisation (WFP). Emeis (2010) introduced a wind deficit model for wind farms, concluding that the wind deficit decreases because of momentum inflow from higher altitudes. In his Lagrangian

approach, an air parcel is described, travelling from the end of the wind farm downstream in the wake for several kilometres. Emeis (2017) presents a time dependent solution for an exponential decrease of the wind deficit. While clustered WECs and their wind deficit have an influence on the meso-scale weather, individual WECs influence the atmospheric flow and turbulent transports on a smaller scale in their respective wake area. A key parameter is to resolve the wind deficit as a main driver of shear stress in the wake. In numerical simulations wind deficit models based on the thrust coefficient of the WEC are common

practice. For example Magnusson and Smedman (1999) describe the wind deficit as a time dependent exponential function. Bastankhah and Porté-Agel (2014) also propose a wind deficit model using the thrust coefficient of the energy converter. They also introduce the wake widening as an additional parameter to obtain a more precise prediction in the far wake. While the model by Bastankhah and Porté-Agel (2014) provides a spatial distribution for the wind deficit, it lacks atmospheric boundary conditions (turbulence, thermal stratification etc.) that have an influence on the wake development. Bastankhah and Porté-Agel

(2017) refined their analytical model in a wind-tunnel, however, atmospheric conditions are hard to simulate in wind-tunnel experiments. Usually, wind-tunnel results suffer from scaling errors when applied to the real world. Yet, for the simulated conditions (thermally neutral stratification and along the dimensions of the wind-tunnel) the model from Bastankhah and Porté-Agel (2017) works fine.

     Hirth and Schroeder (2013) conducted a field measurement, using a mobile Doppler radar system to span a high resolution

grid of radar measurements behind a WEC, studying the wind deficit influence of a neighbouring converter, in order to quantify the power losses in wind farms.

     In the present study in-situ wake measurements are used to display the wind deficit behind an Enercon E-112 converter in the near to mid wake (0.5 to 5 rotor diameters D). In-situ measurements have the advantage that the measured wind is already a superimposition of turbulence created by the WEC and the free stream turbulence. Consequently a model derived from

UAS in-situ data needs to accommodate turbulence and shear terms in some way. Disadvantages of real-world experiments are, of course, additional boundary conditions that are hard to quantify, e.g. is vegetation, additional WECs or obstacles of any kind (dyke, buildings) creating random, unrelated (to the investigated problem) turbulence that adds uncertainties to the measurement. Despite these issues, in the following, a steady state spatial analytical solution for a wind deficit is presented, based on the momentum flux conservation equation.

## 2   Theory and methods

### 2.1   Analytical solution of the conservation of momentum equation

The starting point for the presented analytical wind deficit model is the equation for conservation of momentum in the mean flow using Einstein summation notation (Stull, 1988). After Reynolds averaging and the Boussinesq approximation have been



applied, the conservation of momentum can be written as:

$$\underbrace{\frac{\partial \overline{u_i}}{\partial t}}_{\text{I}} + \underbrace{\overline{u_j}\frac{\partial \overline{u_i}}{\partial x_j}}_{\text{II}} + \underbrace{\delta_{i3}g}_{\text{III}} - \underbrace{f_c\epsilon_{ij3}\overline{u_j}}_{\text{IV}} + \underbrace{\frac{1}{\overline{\rho}}\frac{\partial \overline{p}}{\partial x_i}}_{\text{V}} - \underbrace{\frac{\nu\partial^2\overline{u_i}}{\partial x_j^2}}_{\text{VI}} + \underbrace{\frac{\partial \overline{u_i'u_j'}}{\partial x_j}}_{\text{VII}} = 0 \tag{1}$$

Here, $i,j = 1,2,3$ for all three directions in space, $g$ is the gravitational acceleration, $p$ the static pressure, $f_c$ the Coriolis parameter, $\nu$ the kinematic viscosity and $\rho$ the density of air.

| | |
|---|---|
| Term I | represents storage of mean momentum. |
| Term II | describes advection of mean momentum by the mean wind. |
| Term III | allows gravity to act in the vertical direction only. |
| Term IV | describes the influence of the Coriolis force. |
| Term V | describes the mean pressure-gradient force. |
| Term VI | represents the influence of viscous stress on the mean motions assuming incompressibility. |
| Term VII | represents the influence of Reynolds' stress on the mean motions. |

Term II can be rewritten for an incompressible flow (and thus divergence-free) (Etling, 2008):

$$\overline{u_j}\frac{\partial \overline{u_i}}{\partial x_j} = \frac{\partial}{\partial x_j}(\overline{u_j u_i}) - \overline{u_i}\underbrace{\frac{\partial \overline{u_j}}{\partial x_j}}_{=0} \tag{2}$$

A one dimensional, horizontal steady state wind field with no buoyancy is assumed (no change in wind direction and wind speed at each location and time). No meso-scalic changes in the static pressure occur in the model. Thus, term I, III and term V can be cancelled from the equation. In this study the wake is observed up to $5\,\mathrm{D} \approx 600\,\mathrm{m}$, as a consequence the Coriolis force (term IV) can be neglected. Term VI represents the viscous stress and observations in the atmosphere indicate that the molecular diffusion terms are several order of magnitudes smaller compared to the other terms and can be neglected (Stull, 1988). Equation 1 can now be simplified for $\overline{u} = u_r$, the reduced horizontal wind speed in the wake along the $x$ direction, with the remaining terms II and VII, as:

$$\underbrace{\frac{\partial(u_r \cdot u_r)}{\partial x}}_{A} + \underbrace{\frac{\partial(\overline{v}\cdot u_r)}{\partial y}}_{B} + \underbrace{\frac{\partial(\overline{w}\cdot u_r)}{\partial z}}_{C} + \underbrace{\frac{\partial(\overline{u'u'})}{\partial x}}_{D} + \underbrace{\frac{\partial(\overline{u'v'})}{\partial y}}_{E} + \underbrace{\frac{\partial(\overline{u'w'})}{\partial z}}_{F} = 0 \tag{3}$$

With the $x$-axis rotated into the main wind direction $\overline{v}$ becomes zero. Therefore term B can be cancelled from Eq. 3. For further simplification, only vertical momentum transport is assumed and lateral momentum transport is ignored, due to the absence of change in the lateral shear stress (term D and E can be cancelled). In a homogeneous unstable to neutral boundary layer subsidence can be neglected and therefore $\overline{w} = 0$; term C can be cancelled (Stull, 1988).

$$\frac{\partial u_r^2}{\partial x} + \frac{\partial(\overline{u'w'})}{\partial z} = 0 \tag{4}$$

Equation 1 is now reduced to a much simpler differential equation in one dimension by cancelling insignificant parts, considering the order of magnitude of each term. The resulting Eq. 4 states that a change in the wind speed in $x$ direction corresponds to



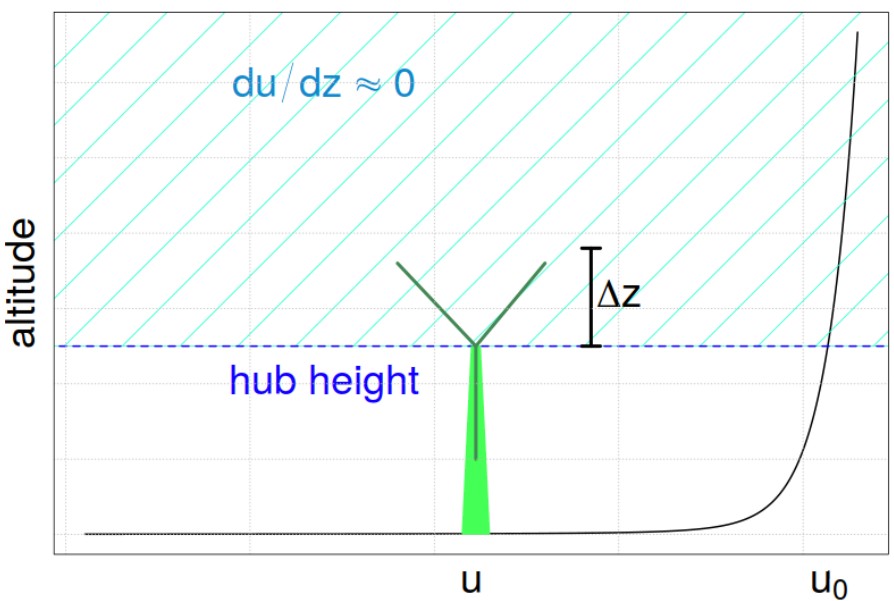

**Figure 1.** Sketch of the wind conditions for a large WEC in an unstable boundary layer. The wind speed does not increase much around hub height $z$, so $u(z) \approx u(z + \Delta z)$.

a change of vertical momentum (in-)flux. This equation is the Eulerian analogy to Eq. 17 in Emeis (2010) using an Lagrangian approach.

The Reynolds shear stress can be expressed using a momentum transfer coefficient $K_{\mathrm{m}}$:

$$\overline{u'w'} = \frac{\tau_{\mathrm{zx}}}{\overline{\rho}} = -K_{\mathrm{m}}\frac{\Delta u}{\Delta z} = -K_{\mathrm{m}}\frac{u_0 - u_r}{\Delta z} \tag{5}$$

5    In analogy to the derivation of a wind deficit model for wind parks by Emeis (2010) a vertical space $\Delta z$ is defined above hub height $z$ within which the wind speed is reduced to $u_r$ and where momentum is transported into the wake from above. Above $\Delta z + z$ an undisturbed flow velocity $u_0$ is assumed which varies only little with height (see Fig. 1). Thus the momentum flux within $\Delta z$ can be approximated by Eq. 5.

For the calculations presented later in this study, $K_{\mathrm{m}}$ needs to be resolved:

10    $$K_{\mathrm{m}} = \kappa \cdot u_* \cdot z \tag{6}$$

With $u_*$ the shear stress velocity, altitude (hub height) $z$ and $\kappa$ the van Kármán constant (Foken, 2006). Regarding the temperature profile, wind conditions and turbulence intensity (s.a Sec. 3.3), a stability parameter $\zeta = z/L$ of approximately 0 to 1 can be concluded, using Businger et al. (1971). Joffre et al. (2001) studied the variability of the stable and unstable atmospheric boundary layer height and documented a dependence of the shear stress velocity $u_*$ on the stability parameter $\zeta$. From there,



a value of $u_* \approx 0.3$ can be concluded for the respective atmospheric stability, present at the time of the field measurements. Slight differences in $u_*$ solely shift the solution along the $y$ axis. In Sec.5 a short sensitivity evaluation is presented.

Now, that the Reynolds shear stress is parametrised, Eq. 4 can be re-written by making the occurring partials to differences and $\Delta u = u_0 - u_r$ the difference between the undisturbed wind speed $u_0$ and the reduces wind speed $u_r$ in the wake, cf. Eq. 18 in Emeis (2010). After inserting Eq. 5 into Eq. 4 the differential equation Eq. 4 becomes:

$$\frac{\Delta u_r^2}{\Delta x} = (u_0 - u_r)\underbrace{\frac{K_\mathrm{m}}{\Delta z^2}}_{\alpha} = \alpha u_0 - \alpha u_r \tag{7}$$

$$\frac{\Delta u_r^2}{\Delta x} = \alpha u_0 - \alpha u_r \iff 2u_r\frac{\Delta u_r}{\Delta x} = \alpha u_0 - \alpha u_r$$

Rearranging and moving $\Delta x$ to the other side

$$\iff \Delta u_r = \left(\frac{\alpha u_0}{2u_r} - \frac{\alpha}{2}\right)\Delta x \tag{8.1}$$

integrating on both sides yields:

$$\iff u_r = \left(\frac{\alpha u_0}{2u_r} - \frac{\alpha}{2}\right)x + c$$

Equation 7 is a non-homogeneous non-linear differential equation (DE) of first order. The solutions of the corresponding homogeneous DE are of the form:

$$u_r^\mathrm{hom} = \pm\sqrt{\alpha\, u_0\, x} \tag{8.2}$$

which actually is vividly realistic: the reduced windspeed $u_r$ increases by the square-root of the horizontal distance $x$ - quite similar to the vertical development of turbulent boundary layer or an internal boundary layer in heterogeneous terrain (Garratt, 1987, 1994; Hanna, 1987). However, a particular solution of the non-homogeneous DE 7 could not be found. In order to find an approximated solution, $u_r$ is treated as a constant on the right-hand side of Eq. 8.1 and the simplified DE is solved in the following. The obtained result agrees sufficiently well with the numerical solution introduced later (s. Sec. 2.2).

A short assessment of the order of magnitude asures that the introduced error remains small. It is assumed that the change of the velocity gradient along $x$ is small (with the velocity gradient in the order of $\frac{u_0 - u_r(0)}{1000\ \mathrm{m}} \approx \frac{7}{1000}\ \mathrm{s}^{-1}$). This complies with Taylor's hypothesis of frozen turbulence, similarly implemented in the model by Emeis (2010, 2017). This will introduce a small error, as afore mentioned, will be shown later in the results, in a comparison with a numerical solution of Eq. 8.1. Using the described simplification, the analytical solution is much more convenient to solve.

Rearranging again and multiplying with $u_r$

$$\iff u_r^2 + \frac{\alpha u_r x}{2} - c u_r - \frac{\alpha u_0 x}{2} = 0$$

rewrite as a quadratic equation

$$\iff u_r^2 + u_r\left(\frac{\alpha x}{2} - c\right) - \frac{\alpha u_0 x}{2} = 0 \tag{8.3}$$





This quadratic equation has two solutions:

$$u_{r_{1,2}} = \frac{1}{2}\left( \left(c - \frac{\alpha x}{2}\right) \pm \sqrt{\left(\frac{\alpha x}{2} - c\right)^2 + 2u_0\alpha x} \right) \tag{9}$$

While only the positive solution is physically relevant, resulting in positive wind speeds, $c$ can now be determined using initial boundary conditions, i.e. for $x = 0$. These conditions can be measured or a theoretical value can be taken, e.g. $u_r(0) \approx 0.3\,u_0$

using Betz law (Betz, 2013). The equation for the reduced wind speed along the wake can then be rewritten as:

$$u_r(x) = \frac{1}{2}\left( \left(c - \frac{\alpha x}{2}\right) + \sqrt{\left(\frac{\alpha x}{2} - c\right)^2 + 2u_0\alpha x} \right), \text{ with } c = 0.3\,u_0 \tag{10}$$

When examining a wind deficit created by a single converter, a closer look on the deficit formation is necessary. The different wake regions can be divided in three areas that, in numerical modelling, need different approaches (s.a. Fig. 2): the near, intermediate and the far wake (Brand et al., 2011). Directly behind the nacelle in the near wake a low pressure region expands

up to $2 - 3$ D until ambient pressure levels are reached again. After this point, up to $10$ D in the intermediate wake, radial flow shear starts to drain turbulence and the wind deficit is beginning to be reduced (Medici and Alfredsson, 2006; Frandsen, 2007). Note that the radial shear stress is represented by the vertical shear in this 2-dimensional approach (Eq. 4). In the intermediate wake, the wind deficit starts to decay and turbulence and momentum flux gradients begin to decrease. In the far wake (10 to 20 D), the wind deficit dissipates. The above mentioned wake regions are not rigid and depend on the mean wind speed, thermal

stratification, surface roughness etc.

In the presented approach, the wind deficit of a single turbine is described by the vertical momentum flux. And since the near wake and the intermediate wake are of interest, a closer look onto the turbulence distribution inside these parts of a wake, is necessary. Frandsen (2007) describes how the wind deficit is reduced by ambient turbulence acting from the wake boundaries (s.a. green arrows in Fig. 2). The wake can now be divided in turbulence created by the wind deficit (s.a. purple volume in Fig.

2) and a mix of atmospheric and wake turbulence that has 'forgotten its origin' in the outer regions of the wake. Hence, in a wake, the area inhabiting turbulence created by the wind deficit would be rotational symmetric volume that slowly becomes smaller as the distance to the nacelle increases (approximately up to $10$ D).

Considering that the deficit decay depends on the distance $x$, the model will need to satisfy for this condition. Therefore, the frequency $\alpha$ (see Eq. 7) is modified to increase with increasing distance from the nacelle in the wake. In practice, this is

implemented by an approximation of the radius of the core wake, that is still untouched by the free-stream turbulence (s.a. Eq. 11) and is shown by the purple volume in Fig 2. For the one-dimensional approach in this study, this radius $r(x)$ of this rotational symmetric volume can be relabelled to the height $\Delta z(x)$:

$$r(x) = \frac{4R^2}{x + 4R} \overset{1\,D}{=} \Delta z(x) \tag{11}$$

For this study, with the available data, the distance at which the free-stream turbulence caves into the wake is approximately two

rotor diameters ($2$ D $= 4$ R) (Medici and Alfredsson, 2006; Frandsen, 2007). Conclusively, a new function $\Delta z(x)$ is needed. Considering all requirements to the new relation, Eq. 14 is formulated. This equation is not continuously differentiable, hence





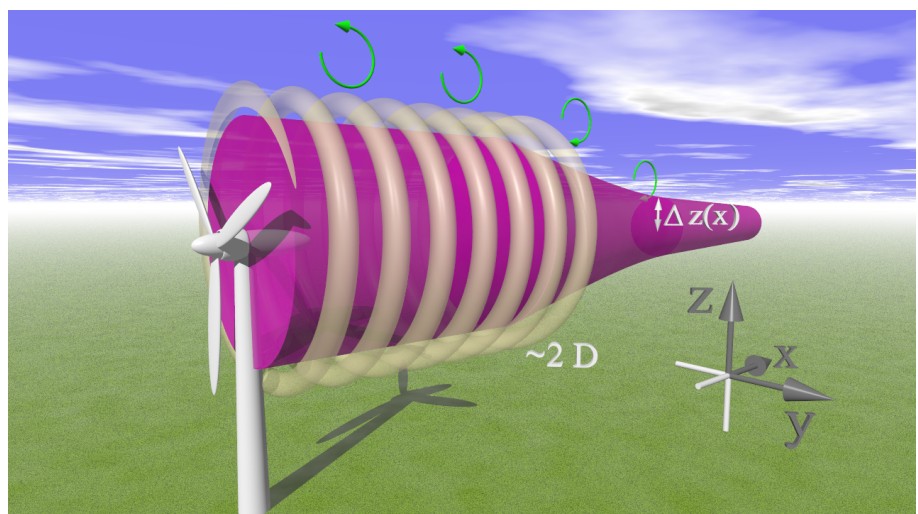

**Figure 2.** Sketch of a WEC wake and the wake turbulence development (not true to scale). The coherent helical tip vortex structures (beige) prevents atmospheric turbulence (green arrows) to enter the wake up to about 2 D. After this point, the helical tip vortex structure begins to decay and atmospheric turbulence (green arrows) can enter the wake. The area inhabiting turbulence 'knowing its origin' from the WEC thins out with increasing distance to the nacelle (purple volume).

the kink in its analytical solution (Fig. 3). In the real world this boundary would be fluent (cf. the numerical solution in Fig. 8). The 'dynamic alpha' analytical solution is computed in a semi-numerical way, this means that $\alpha(x)$ is calculated using Eq. 12 for each distance and then inserted into Eq. 10. A sole analytical solution is possible inserting Eq. 14 into Eq. 8.1. However, for this proof of concept with the Euler method as validation (s.a. Sec. 2.2), the more convenient and easy to implement method of

5     the semi-numerical solution is used.

$$\alpha(x) = \frac{K_{\mathrm{m}}}{\Delta z(x)^2} \tag{12}$$

| $\alpha = \text{constant}$ | $\alpha = \text{dynamic}$ |
|---|---|
| $\Delta z(x) = R \hspace{2em} (13)$ | $\Delta z(x) = \begin{cases} R & \text{if } x \leqq 4R \\ \frac{4R^2}{x+4R} & \text{if } x > 4R \end{cases} \hspace{1em} (14)$ |

## 2.2 Euler method

The Euler method is a simple way to solve differential equations of first order numerically (Atkinson et al., 2009; Faragó,

10     2013). Thus, the this method solves the presented Eq 8.1 numerically. :

$$u(x)' = f(u(x)) \tag{15}$$

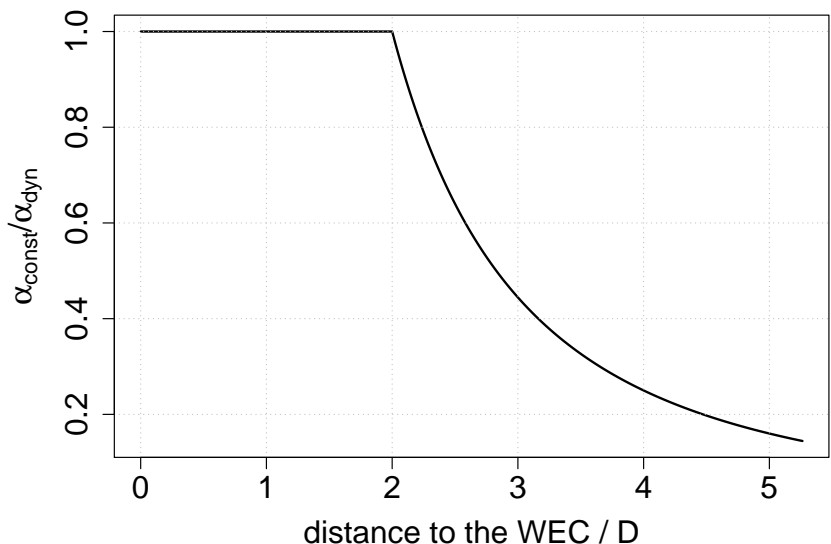

**Figure 3.** A variable $\Delta z(x)$ leading to increasing $\alpha$ values with increasing distance in the wake. A coherent tip vortex helix shields the wind deficit from free-stream turbulence caving in, until at approximately $2\,\mathrm{D}$ wake turbulence and atmospheric turbulence interfere.

The dash denotes a continuously differentiable derivation of a $u(x)$. Discretising each calculation step with:

$$x_n = x_0 + h\,n, \text{ with } n = 1, 2, 3, ... \tag{16}$$

The differential form is approximated by a discrete expression, with a discretisation step $h = 0.1\,\mathrm{m}$ in the presented study.

$$\frac{u(x_{n+1}) - u(x_n)}{h} \approx f(u(x_n)) \;\Rightarrow\; u(x_{n+1}) \approx u(x_n) + h\,f(u(x_n)) \tag{17}$$

5 Reducing $h$ further does not change the result, but increases the computation time. Thus, the step size $h = 0.1\,\mathrm{m}$ is used for the numerical solutions. Both, the forward and backward method, are used to validate the analytical solution and to obtain a numerical solution for the difference Eq. 8.1, a short derivation of the specific Euler solutions is shown below.

Inserting Eq. 8.1 in Eq. 17 with $\Delta x = h$ and $u = u_r$ the Euler forward solution is:

$$u(x_{n+1}) \approx u(x_n) + h\left(\frac{\alpha u_0}{2u(x_n)} - \frac{\alpha}{2}\right) \tag{18}$$

10 While Eq. 17 is called the Euler forward or explicit method, also an Euler backward or implicit method (Eq. 19) is calculated, where the solution is only available implicitly, hence the name. While the Euler forward method is straight forward and simple to solve, for the Euler backward solution Eq. 19 has to be solved for $u(x_{n+1})$. Note the appearance of $u(x_{n+1})$ on both sides of the equation.

$$u(x_{n+1}) \approx u(x_n) + h\,f(u(x_{n+1})) \tag{19}$$





Re-writing Eq. 18 for the backward solution method yields:

$$u(x_{n+1}) \approx u(x_n) + h\left(\frac{\alpha\,u_0}{2\,u(x_{n+1})} - \frac{\alpha}{2}\right) \tag{20.1}$$

$$\approx u(x_n) + \frac{h\alpha\,u_0}{2u(x_{n+1})} - \frac{h\alpha}{2} \tag{20.2}$$

Multiplying both sides of the equation with $u(x_{n+1})$

$$\Longleftrightarrow u(x_{n+1})^2 \approx u(x_{n+1}) \cdot u(x_n) + \frac{h\alpha\,u_0}{2} - \frac{h\alpha\,u(x_{n+1})}{2} \tag{20.3}$$

Rearranging to a quadratic equation gives:

$$\Longleftrightarrow u(x_{n+1})^2 - u(x_{n+1})\left(u(x_n) - \frac{h\alpha}{2}\right) - \frac{h\alpha\,u_0}{2} \approx 0 \tag{20.4}$$

The resulting specific Euler backward solution is the positive version of the solution of the quadratic Eq. 20.4:

$$u(x_{n+1}) = \frac{u(x_n) - \frac{h\alpha}{2} \pm \sqrt{\left(u(x_n) - \frac{h\alpha}{2}\right)^2 + 2\alpha u_0 h}}{2} \tag{21}$$

## 3 Data acquisition and data availability

### 3.1 Measurement system

All data were captured using the in-house developed MASC (multi-purpose airborne sensor carrier) small UAS of the University of Tübingen. With a wingspan of roughly $4\,\mathrm{m}$ it can carry a payload of $2.5\,\mathrm{kg}$ and fly autonomously up to 2 hours using the Pixhawk 2.1 autopilot (Pixhawk-Organisation, 2019), propelled by an electrical pusher motor. MASC is equipped with a five-hole probe to resolve turbulent pressure fluctuations, GPS and a platinum fine wire resistance thermometer at the nose of the UAS (s.a. Fig. 4). With its set-up it is possible to calculate the 3-D wind vector at any point in space up to 30 Hz. A more detailed description of the UAS and its instrumentation can be found in Rautenberg et al. (2019b) or Wildmann et al. (2013, 2014a, b).

### 3.2 Experimental site

The underlying in-situ data were captured behind an Enercon E-112 converter that is part of the Jade wind park north of Wilhelmshaven in Germany. The WEC is located roughly $2\,\mathrm{km}$ from the North Sea coast line. In mid November 2019 a measurement campaign took place for the HeliOW (Helicopter flights in Off-shore Wind parks) project funded by the BMWi (Federal Ministry of Economic Affairs and Energy in Germany). The project aims to determine safe helicopter corridors in offshore wind parks, since wind parks are getting larger and single WECs continue to increase in dimensions. The wind measurements form the foundation for a numerical simulation chain that investigates the impact of blade tip vortices and wake turbulence on helicopter flight dynamics (Horvat et al., 2020) from numerical simulated wake data (Cormier et al., 2018), accompanied with piloted simulations using the AVES helicopter flight simulator at the German Aerospace Center (Strbac





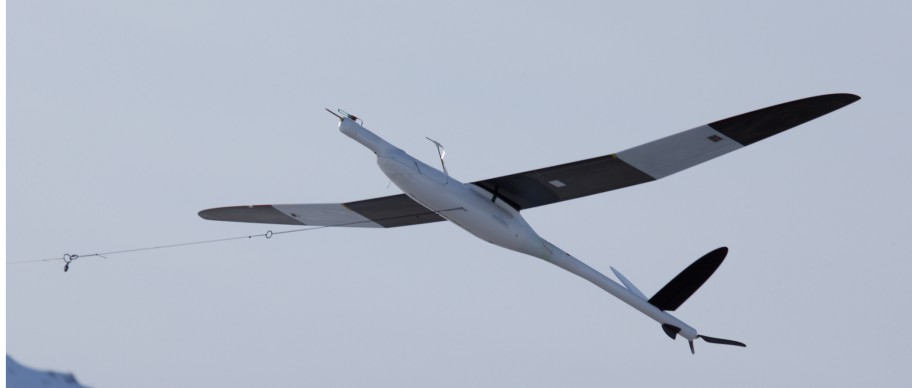

**Figure 4.** In-house built small UAS of type MASC of the University of Tübingen shortly after lift-off. At the front of the UAS the five hole probe and the fine wire resistor thermometer are visible. The air plane is propelled by an electrical pusher motor.

et al., 2019). At a previous campaign in 2018, measurement strategies were tailored toward measuring blade tip vortices (Mauz et al., 2019); whereas in the recent campaign the goal was to cover the wind deficit in the near to intermediate wake at hub height. To accomplish this, a meander pattern was flown by the UAS (unmanned airborne system) covering the wake from 0.5 D to 5 D (Fig. 5), legal restrictions prevented farther downstream reaching measurements. Altogether, three consecutive flights

are available for different flight legs behind the WEC. A leg is the straight and level flight path in a measurement flight pattern. But as it turned out, in flight 1 the legs were too short to cover the whole wake downstream, thus in flight 1 the number of measurement legs is reduced to the ones in the near wake. Due to high wind speeds the UAS needed a larger turning circle as usual and the flight pattern needed adjustment (s.a green tracks in Fig. 5).

### 3.3 Atmospheric conditions

The measurement flights took place November 15 2019 at 15:30 LT in the Jade wind park with an easterly wind direction. An average wind velocity of $10.5 \mathrm{~m~s^{-1}}$ in hub height was measured. The turbulence intensity at this altitude was about $5-8$ %. Figure 6 shows a vertical profile of the virtual potential temperature in the inflow that suggests near neutral conditions above 20 m a.g.l.

### 4    Results

In this section the MASC measurements are compared to two analytical solutions (one with constant and one with dynamic $\alpha$ as seen in Eq. 12) and the numerical solutions (Eq. 17 and Eq. 21) of the simplified momentum flux conservation equation. It can be shown that Euler forward and backward method are identical. Both introduce a small error that increases with increasing $\Delta x$. However, the analytical solution Eq. 10 is not far off from the numerical solution. In the presented data the maximum deviation between the analytical and numerical solution is $\Delta_{\max} \approx 5$ % at 5 D. An additionally occurring parameters in the





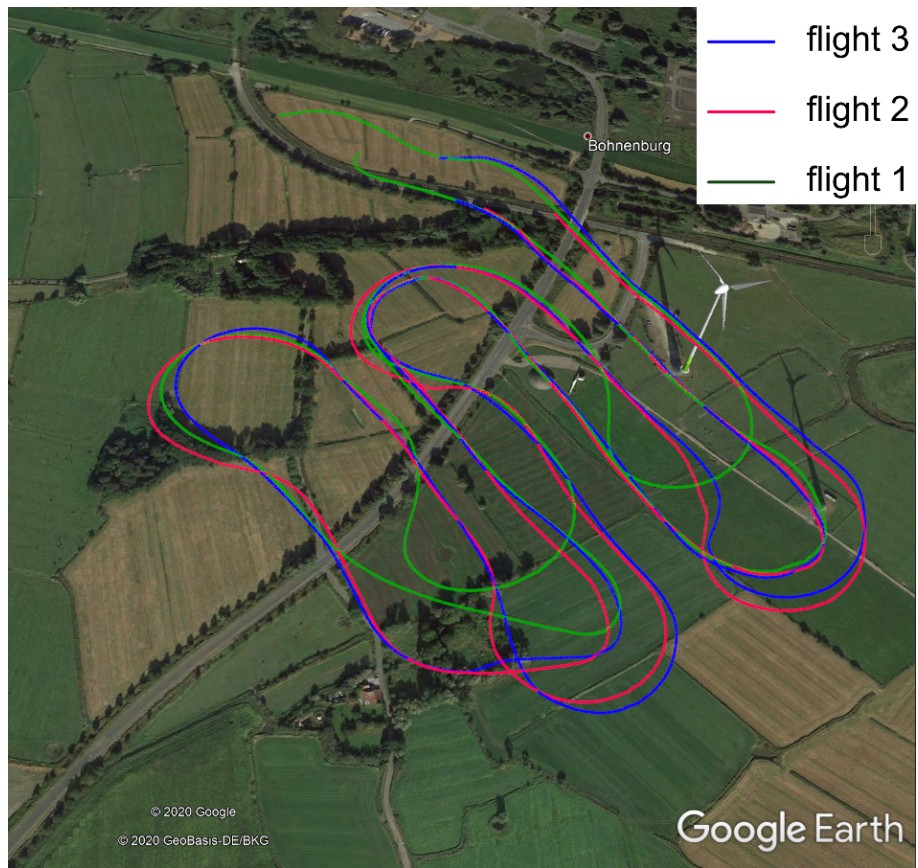

**Figure 5.** Flight paths of the MASC-3 UAS in the wake behind the E-112 WEC. The measurement legs of flight 1 (green line) were too short, thus the legs were extended for the second and third flight (red and blue line). Therefore, limited data are available for flight 1.

equations is the hub height of the E-112 converter. This hub height is set to $z = 125$ m. In the $\alpha = $ const. calculations, $\Delta z(x)$ is set to half a rotor diameter throughout the calculations.

### 4.1 Solutions using a constant $\alpha$

The assumption of a constant $\alpha$ is the simplest approach, but neglects the measurements of Medici and Alfredsson (2006) and the theory by Frandsen (2007) who studied i.a. turbulence entrainment in the near wake. Figure 7 shows the analytical model in a black solid line, both Euler series in red and purple dashed lines with the underlying MASC in-situ data. The analytical and numerical solution fit the data well up to the point where free-stream turbulence has an influence on to wind deficit. From the measured data, this distance can be derived to be about $2 - 3$ D $\approx 230 - 300$ m. From this point both calculated solutions seem to no longer represent the measured data. This is also the region in which the helical tip vortex structure has collapsed, lost its rotational momentum and can no longer shield the wind deficit from the surrounding flow turbulence and shear. This

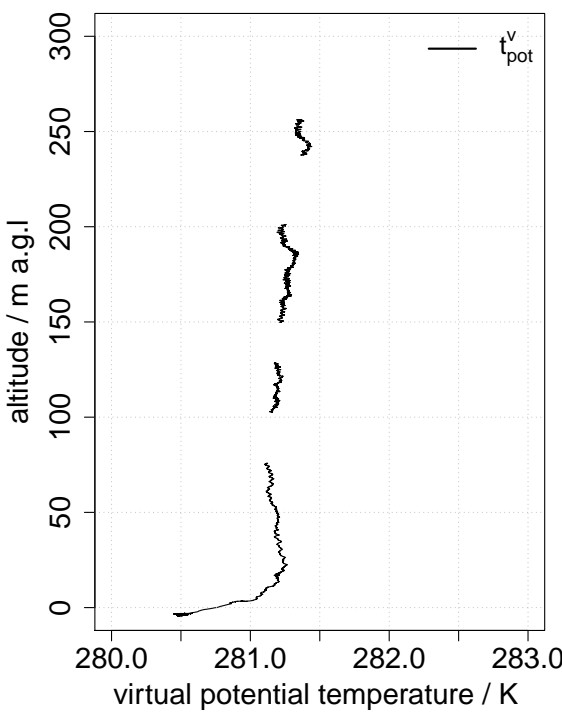

**Figure 6.** Vertical profile of the potential virtual temperature at about 2 D in front of the WEC. Measurements obtained in curves are clipped from the graph, hence the three data lags in the profile.

region also depends on the tip speed ratio and therefore is variable and coupled to the wind conditions and the operation of the WEC (Porté-Agel et al., 2020).

## 4.2 Solutions using a dynamic $\alpha$

To represent the MASC data and to do justice to the entraining turbulence accelerating the wind deficit decrease, Eq. 10 and

5 Eq. 8.1 need a slight modification. Now, Eq. 14 is used, thus, $\alpha$ defined in Eq. 12, is now a function of $x$. As described above, $\Delta z$ corresponds to the radius in the wake of 'pure wake turbulence', still knowing its origin. With this parameter decreasing downwind of the wake, $\alpha$ increases along the $x$ axis. This essentially creates a high turbulent region in which the turbulent momentum flux from aloft increases the decay of the wind deficit and conversely increases the mean horizontal wind. Figure 8 shows the result of an increasing $\alpha$ using $\Delta z(x)$ defined by Eq. 14 and shown in Fig. 3. The analytical and

10 the numerical solution now follow the measured data. The measured data scatter about $\pm 5$ % around the computed solutions.

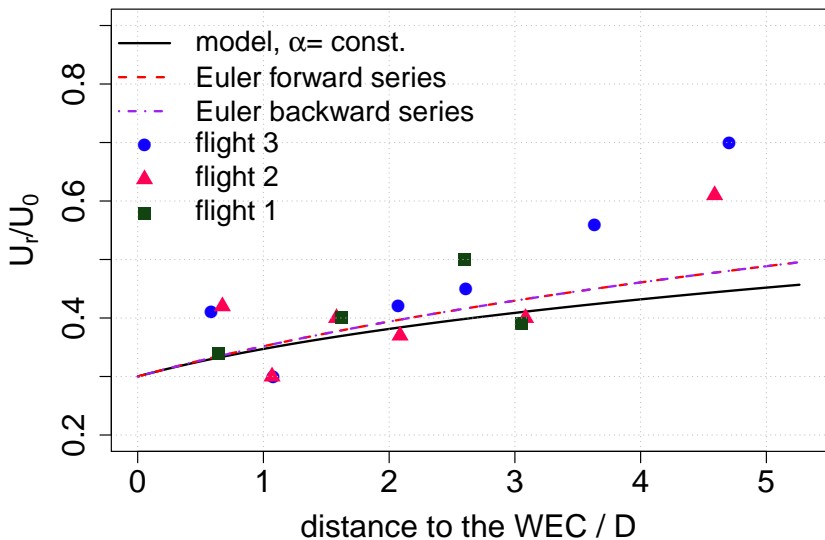

**Figure 7.** Wind deficit measurements of three flights at hub height behind an Enercon E-112 WEC. The red and purple dashed lines are the numerical solutions of Eq. 8.1. The analytical (black solid line) and numerical model use a constant $\alpha$ along $x$. A deviation between the analytical and numerical solution of 5 % at 5 D can be measured. At about 2.5 D the measurements follow a different trend than the computed solutions.

The most important feature is the kink at $x = 2$ D, where both solutions are adjusted by considering the entraining free-stream turbulence, following the progressively decreasing wind deficit.

### 4.3 Far wake behaviour of the model

In the sections above the behaviour of the models is shown up to 5 D from the WEC. The far wake, roughly beginning at
5   $8 - 10$ D and beyond, is characterised by 'uniform' turbulence, meaning that atmospheric turbulence and turbulence created by the WEC are undistinguishable from one-another. The additionally created turbulence by the WEC has almost dissipated and the wake stream is reintegrated into the surrounding flow. Figures 9 and 10 show the behaviour of the models with constant and dynamic $\alpha$ for the far wake. While the constant-$\alpha$ model underestimates the wake behaviour the dynamic-$\alpha$ approach follows the measured data up to 5 D and paints a reasonable picture of the wind deficit decay. Eventually, the numerical solution runs
10   into saturation at $u_r/u_0 = 1.0$ at about 12 D.

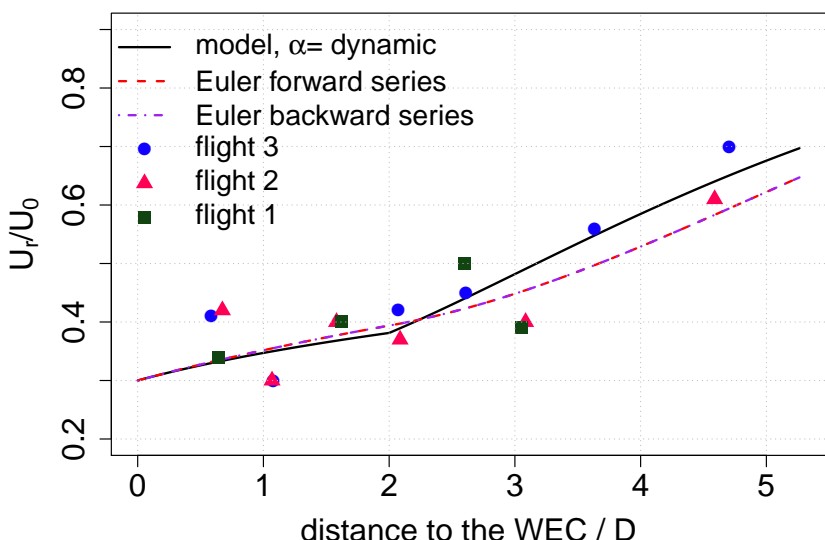

**Figure 8.** Wind deficit measurements of three flights at hub height behind an Enercon E-112 WEC. The analytical and numerical model use a dynamic $\alpha$ to satisfy for a decreasing $\Delta z$ for increasing $x$. In direct comparison to Fig. 7 it can be seen that the modelled solutions now follow the measured data. By the implementation of a dynamic $\alpha$ both solutions satisfy for the change in turbulence entrainment at around 2.5 D.

## 5 Discussion

The presented analytical and numerical solution acknowledge the near-wake turbulence behaviour. As shown above, the adjusted model (using a dynamic $\alpha$) follows the measured data. In this first approach a rather simple hyperbolic function (Eq. 14) has been used to describe the WEC turbulence length that is not continuously differentiable, but there might be a smoother

5   function that simulates the same behaviour.

The presented model can further be understood as a simple solution of the turbulent momentum conservation equation for a specific initial condition. The resulting equations are the solution of the simplified momentum flux conservation for the atmospheric flow that has been brought out of its equilibrium. The presented solutions do not represent the shear stress that is created by the WEC itself (solely by its presence in the flow). The solutions simulate the surrounding atmosphere refilling

10   a 'velocity gap', created by a horizontal wind depression in flow direction ($u_r(0) = 0.3 \, u_0$). The shear stress created by the presence of the WEC, is automatically dealt with by the behaviour (steepness, trend) of $\Delta z(x)$.

The shear velocity $u_*$ is usually measured at the ground. In this field campaign no ground station was available. Thus, a value for $u_*$ had to be derived from the thermal stratification of the lower atmosphere. Although the MOST studies are widely



WIND
ENERGY
SCIENCE
DISCUSSIONS

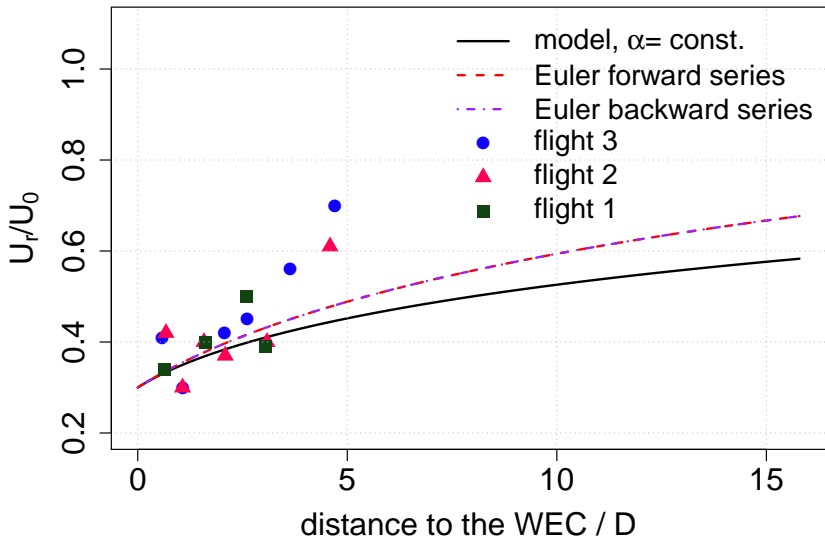

**Figure 9.** Same as Fig. 7 but extended to the far wake behaviour of the analytical and numerical model. The far wake is underestimated by $\approx 0.1$ at 15 D in the analytical solution.

accepted, a derivation of $u_*$ from thermal stability always contains uncertainties. Therefore, the influence of the shear velocity $u_*$ has been examined by changing the input value by $\pm 50\%$. The resulting analytical solution then deviates by a maximum of approximately $\pm 5\%$ at 5 D in the resulting wind reduction, from the solution shown in this study. Thus, for example a $u_*$ of 0.45 still results in a good agreement with the data, while still be representative for a neutral to slightly unstable atmosphere
5  (Businger et al., 1971).

## 6 Conclusions

The derived analytical solution shows good agreement with the in-situ UAS measurements. It is shown that in the wake, considering the free-stream turbulence distribution, and its inability to affect the wind deficit unevenly in the wake, is important for the development of the wind deficit. Although the lack of data for the far wake, the dynamic-$\alpha$ model connects the near
10  wake with the intermediate and far wake turbulence and wind deficit. In this study a literature value for $u_*$ is used. In a future measurement campaign, the shear stress velocity could be derived from two measurement legs in front of the WEC in different altitudes, taking advantage of the logarithmic wind profile. A direct measurement of the Reynolds stress $\overline{u'w'}$ at hub height could also be feasible. Also the wake development farther down the wake would be interesting to cover in future measurements, e.g. up to $10 - 15$ D.

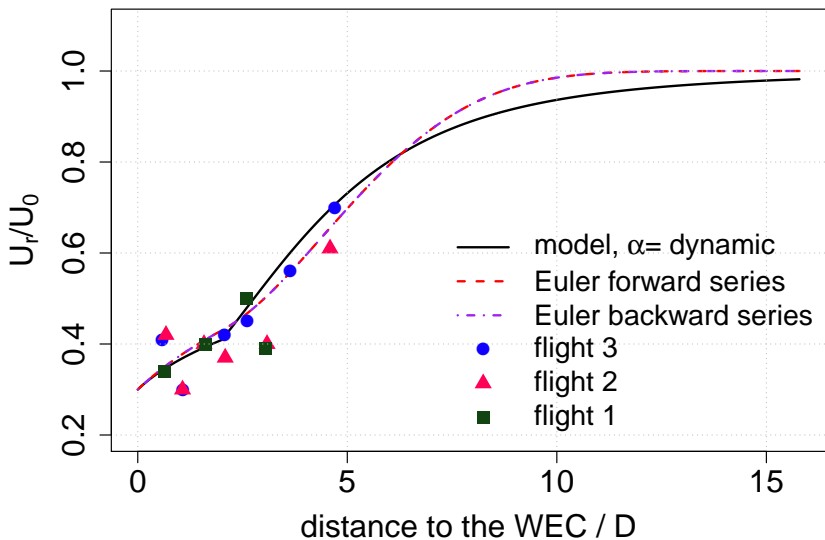

**Figure 10.** Same as Fig. 8 but extended to the far wake behaviour of the analytical and numerical model. The free stream velocity of $0.95 \cdot u_0$ in the wake is reached at ca. $10 - 12$ D. Also the numerical model reaches its saturation of $u_r/u_0 = 1.0$ at around 12 D.

Regarding the far wake behaviour of the dynamic-$\alpha$ approach, it could be shown, that the analytical and numerical solution can predict an internal boundary layer. One of the main features of this approach (e.g. for an internal boundary layer model) is the capability to reach a certain saturation level. This can be a great advantage in modelling wind transitions from land to sea (or vice versa) or thermal internal boundary layers (TIBL) in general.

5   The model presented in this case can also be applied to larger scale problems. For example to describe the wind deficit of a wind farm. Another interesting case is the additional consideration of the lateral turbulent momentum flux that must occur, as soon as the wind deficit disturbs the idealised neutral conditions of the free atmospheric flow (term E in Eq. 3).

*Data availability.* The measured UAS data can be provided by the authors (Moritz Mauz, Jens Bange, Andreas Platis) by ftp. Due to the amount and complexity of the data, we advise a brief introduction by one of the authors.



# Appendix A

## A1

*Author contributions.* MM evaluated the UAS data and derived the formalism. BvK supported the methods with his expertise and his ABL knowledge. AP added to the discussion and understanding of the topic. SE and JB shaped the manuscript and added valuable input to form a common theme. All authors added constructive criticism during the process of writing the manuscript.

*Competing interests.* The authors declare that they do not have competing interests.

*Acknowledgements.* Financial support: This research has been supported by the Projektträger Jülich, the BMWi (Federal Ministry for Economic Affairs and Energy) that funded the HeliOW project (0324121C).



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
