# Peer review of "An analytical solution for wind deficit decay behind a wind energy converter using momentum flux conservation validated by UAS data"

_Wind Energy Science, 2020_

## Referee Comment (RC1) · Anonymous Referee #1 · 5 Aug 2020

The authors present an equation to reproduce the wind deficit behind a wind turbine, and compare their results with unmanned aerial system (UAS) observations.

I appreciate the general strategy of trying to establish a wake thickness description with a model based on momentum equation and checked against UAS observations. However I do not find the paper really convincing neither in the analytical modelling nor in the observational section. The observations presented in the paper are restricted to a few values of the mean wind measured by the UAS at several distances from the rotor. Furthermore, some key parameters, such as the friction velocity used by the authors in their parameterization, are not computed from observations (or retrieved from

simulations with a weather model), but arbitrarily fixed to a value of 0.3 m/s, supposed representative of a wide range of meteorological conditions. The mathematical developments are confusing, and, in my opinion, wrong in some parts. I find it hard to believe that all of the coauthors have carefully examined this manuscript before its submission to WES.

There is abundant literature on turbine wake observation and modeling (the authors mention the comprehensive review paper by Porté-Agel et al., which appeared this year in Boundary-Layer Meteorology). It is therefore crucial that any new article clearly explains what is brought with respect to the existing knowledge. To summarize, the present manuscript needs a lot of work, on both form and content, before becoming acceptable in the Journal.

Major comments:

1. I do not find any interest in the Euler method to solve the model equation. If there exist an analytical solution, why playing with approximate, numerical solving? This adds confusion.

2. A lot of analytical models describing the wind deficit behind a rotor are already available in the literature. The authors do not explain why there is a necessity for a new one, what is the improvement brought by their model, how it compares with the existing ones, etc.

3. The hypotheses used in the equations lead to a mathematical impasse: it is assumed that the transversal and vertical wind components are zero ($<v>=<w>=0$; I use brackets here instead of overbars for easy writing). We thus have $d<v>/dy= d<w>/dz=0$, and to satisfy incompressibility in mass conservation equation, we therefore get $d/dx=0$!. Furthermore, the authors come to the relation $d/dx=d/dx$ (equations (2) and (4)), in contradiction to the mathematical relation $d/dx = 2d/dx$.

4. The authors mix partial derivative equations and bulk approximations with finite differences (e.g. equation (5)). If an analytical solution is to be found, then the mathematical developments have to be conducted with the derivative forms (i.e. not approximate) of the equations.

5. The manuscript reveals weaknesses in the knowledge of boundary-layer meteorology. It is mentioned that the model is applied above the surface layer ($d/dz=0$ from the hub height), but the equation used to compute the eddy-diffusivity (eq. 6) is valid in the surface layer (and for neutral conditions). A sentence such as "Regarding the temperature profile, wind conditions and turbulence intensity (s.a Sec. 3.3), a stability parameter $\ddot{I}\check{Z} = z/L$ of approximately 0 to 1 can be concluded, using Businger et al. (1971)." is really annoying, because estimating the stability requires the knowledge of friction velocity and buoyancy flux, and none of these two parameters was measured during the observation periods.

6. The results presented in Figs. 7 to 10 should be grouped in a single graph (note that the observations presented in these 4 figures are the same). The curves relative to Euler's solutions should be omitted, as well as the model curves which are not relevant ($\alpha$=constant for distances to the turbine larger than two diameters).

Specific comments and technical errors:

1. There is no mention of the turbine parameters (e.g. the thrust coefficient), though some wake models involve such parameters in their equations. This should be commented.

2. P. 2, L. 19-21: Is there any specific interest to mention this study in regard to numerous other observations done in the wake of wind turbines?

3. P. 3, eq. (2): in the rhs, uj and ui should be overlined separately (i.e., with bracket notation,  instead of ).

4. P. 3, L. 10-12: If there is no pressure gradient, then there is no geostrophic wind,

and in non-perturbed conditions the wind comes to zero. A geostrophic balance (compensation between pressure and Coriolis forces) should instead be invoked here.

5. P. 3, lines 19-20: It should be explained why subsidence might be neglected in unstable and neutral conditions (implying a different behaviour in stable conditions?).

6. Fig. 1: The wind profile represents the conditions ahead of the wind turbine. A second profile representative of the wake conditions should be added.

7. To avoid confusion, I suggest to replace z (hub height) and $\Delta z$ with, e.g. h and $\Delta h$.

8. P. 4, L. 11: Typo "von Karman".

9. P. 4, last line: "Joffre et al. (2001) studied the variability of the stable and unstable atmospheric boundary layer height and documented a dependence of the shear stress velocity u* on the stability parameter ÏŽ." The main driver of u* is the wind. Furthermore, u* is one of the two parameters used to define the stability (and not the other way).

10. P. 5, L. 2: "Slight differences in u* solely shift the solution along the y axis". Please explain. What is "the y-axis"?

11. P. 5, L. 3: Unclear for me.

12. P. 5, L. 4: Typo "reduced".

13. P. 5, eq. 7: Please define what $\alpha$ represents.

14. P. 5: "Equation 7 is a non-homogeneous non-linear differential equation (DE) of first order". As it stands, Eq. 7 is not a differential equation.

15. P. 5, eq. 8.2: Please define what the superscript hom represents.

16. P. 5: The paragraph "A short assessment ... convenient to solve." Is unclear. Please rephrase.

17. P. 6, eq. 9 and 10: It is not useful to write two equations here.

18. P. 6, L. 24: "the frequency $\alpha$". Why is $\alpha$ called a "frequency"?

19. P. 6, eq. 11: Is there any justification (e.g. a reference) for this equation?

20. P. 6, eq. 11: There is no need to introduce a new symbol (R), since D=2R. Please rewrite as a function of D.

21. P. 7, L. 10: typo "... the this method".

22. Fig. 3: Please define clearly which parameter is represented here.

23. P. 10, L. 3: UAS should be defined at its first appearance (p. 2).

24. P. 10, L. 5: What are the heights of the legs?

25. P. 10, L. 11: Please explain how the turbulence intensity is computed (turbulence observations are not mentioned in the manuscript).

26. P. 10, L. 19: Typo "parameters".

27. Fig. 5: Please add a scale and indicate the geographical orientation. We can observe close to the right border of this image the shadow of a second wind converter. Is there a potential impact of this 2nd converter on the wake of the 1st one?

28. Fig. 6: There are negative altitudes. Please explain.

29. Fig. 6: Please explain why temperature data are discarded during UAS turns. Is that because the measures are biased, or because turns are too far away from the profile location?

30. Fig. 7, caption: replace "At about 2.5 D..." with "From about 2.5 D..."

31. Section 4.3: There are no observations in the far wake area. So, the model performance cannot be evaluated. Why do not try to test the model against another data set?

32. Section 4.3: The sentence "While the constant-$\alpha$ model underestimates the wake

behaviour the dynamic-$\alpha$ approach follows the measured data up to 5 D and paints a reasonable picture of the wind deficit decay." is not relevant for this section.

33. Figs. 7 to 10: The parameter represented is not the "wind deficit".

34. P. 14, L. 3-5: I do not understand what is meant here. Please rephrase.

35. P. 15, L. 4: "0.45 m/s".

36. P. 15, L. 1 to 5: This is surprising: it is known that the greater the turbulence level, the shorter the wind recovery distance in the wake. Furthermore, if u* is a key parameter in the eddy-diffusivity value, then enhancing or lowering it by 50% should significantly modify the wake characteristics.

37. Fig. 10: The curve corresponding to the analytical model here is not identical to that presented in Fig. 8. For example, at a distance of a little less than 5D, the model crosses the blue disk of the observations in Fig. 10, whereas it remains well below in Fig. 8. Please explain.

---

## Referee Comment (RC2) · Anonymous Referee #2 · 6 Aug 2020

Summary

In the manuscript, the derivation and validation of a model for the velocity deficit in the wake of a wind turbine is presented. The model derivation starts from the Reynolds decomposition of the differential momentum equilibrium in a fluid and models a momentum flux from the wind at greater heights, which finally compensates the wake velocity deficit at a certain stream-wise distance to the wind turbine. A differential equation is obtained from the derivation and is solved analytically as well as numerically, where the analytical solution could only be obtained by introducing a simplification. Measurements of the mean wind speed in the wake using an UAV were undertaken to provide

validation data to the derived velocity deficit model. The UAV was equipped with a five-hole probe for the velocity measurement. A flight pattern with 8 horizontal lines parallel to the rotor plane in different distances up to 5D from the rotor was chosen and repeated 3 times. The analytical as well as the numerical solutions of the derived differential equation was compared to the (mean) wake velocities obtained from the measurements. Good agreement was stated up to a distance of 2-3D behind the rotor. After this, the authors claim that the helical tip vortex structure has collapsed and therefore a modification of the derived velocity deficit model is presented. This modification is based on the assumption that a stronger mixing of the wake and the surrounding wind field is apparent from this distance. The modification of the model yields results that better fit the experimental data at higher distances. A discussion on the influence of the shear velocity, which is used as an input parameter of the velocity deficit, is added. In the conclusion, it is stated that the modelled and measured velocity deficit in the wake fit well and a number of possible improvements and further applications of the model are listed.

Comments

Before starting with the detailed comments, one major issue needs to be addressed: The variable $u\_r$ is defined as "the reduced horizontal wind speed in the wake along the x direction". This definition is not sufficient. I assume that $u\_r$ is the mean value of the wake velocity at hub height. All my comments are based on this assumption. Furthermore, it is not clear if the averaging length is one rotor diameter or if the wake expansion is considered (resulting in an increasing length of the averaging space with higher distances from the rotor). Applying the above assumed definition of $u\_r$, the analytical model in Figure 7 shows a reduction of the wind speed in the wake of 70% at 1 D behind the rotor. This is within the scatter of the measurements. This seems to me a surprisingly low mean axial velocity in the wake for a normal operation of the rotor. In wind tunnel measurements of Bartl et al. we see a deficit of 40-50% at that point. Other wind tunnel measurements of Kim et al. show a similar picture at ∼1.5

D with a deficit of a bit more then 40%, while the derived model shows a deficit of more than 60%. PIV measurements performed during the MEXICO experiment also show a considerably lower velocity deficit at design TSR (see Parra et al.). Especially when considering that a higher velocity deficit would be expected due to the absence of atmospheric turbulence in the wind tunnel experiments, the observed and calculated velocity deficit in this work seem surprising to me. From my point of view, it needs to be clarified if this is really the case or if there is a misunderstanding on my side. If not, a discussion on this discrepancy is necessary.

The general idea of the manuscript and the measurements seem promising to me, but the implementation and description of the performed work lacks accuracy at some points, which makes it difficult to judge on the results.

The comments will be clustered in three groups, namely: Derivation of the analytical velocity deficit model, Measurements, General comments.

Derivation of the analytical velocity deficit model

The derivation starts promising with a description of the Reynolds decomposition of the differential momentum equilibrium in a fluid. However, the equation is dramatically reduced by a number of assumptions. After this, the remaining (u'w') term is shall be replaced by an empirical relation. Here, the derivation starts to become difficult to understand and seems to contain some mathematical mistakes or some steps of the derivation were skipped, which prevents the reader from understanding what exactly happened here.

The reduction of the momentum equation is based on several assumptions. The assumption of a 'one dimensional, horizontal steady state wind field' implies that the wind turbine wake is no longer seen as a three dimensional tube or something similar. The model therefore assumes that a momentum flux can only be added to the wake region from higher altitudes but not from the flow on the left and the right from the wake. This assumption is valid for the far wake of wind farms, where the velocity deficits of multiple
wind turbines merge and a more or less homogeneous horizontal layer with a velocity deficit up to a small height (in the order of magnitude of the wind turbine height) in comparison to its lateral size (in the order of magnitude of the wind park width) can be assumed. Here, the influence of the added momentum from the sides is negligible. This is not the case for a single wind turbine and no explanation why this assumption should be valid was found. In addition to that, the authors apply this assumption to the near and mid wake region, which is a region, where the flow is strongly dominated by the geometry of the tip vortex structure. These vortex structures seem completely neglected in this approach.

After reducing the momentum equations, the term (u'w') in EQ 4 is replaced by an empirical correlation, which is inspired by the work of Emeis. (u'w') is set to a term stated by Emeis that models the momentum flux from the above air layers into the wake. In Emeis work, this term is used to compute the integral (from free-stream to hub height) momentum flux. However, EQ 4 is derived from the momentum equilibrium in its differential form, meaning that no integration over the height took place. It is not clear, why this should be valid. This problem is also visible, when differentiating (u'w') by the z coordinate in EQ 7. From my understanding of the derivation, this is simply done by dividing the equation by delta z. Delta z is is defined as vertical the distance of the hub to a flow layer, where no velocity deficit is present. I could not figure out, how the differentiation of the expression in EQ 5 representing the integral momentum flux over the height can lead to this expression. Furthermore, it seems that EQ 7 shows a difference quotient instead of a derivative, which requires a solid explanation. In addition to that, the function shown in EQ 7 seems to be independent from the height, as delta z is a constant as described in line 5, page 4., while EQ 4 is not defined for a certain height. It therefore needs to be clarified if EQ 7 should be an evaluation of EQ 4 at a certain height (including an explanation why this is done).

In EQ 8.1 an integration is performed after rearranging the delta x to the right side. Here, it is still not clear if (delta u_r / delta x) is a derivative or a difference quotient. It

is stated that both sides of the equation will be integrated, but the integration variable is not known. Assuming that x is the variable to integrate over, the dependence of u_r in the denominator of the first term in the braces seems to be ignored.

At that point, so many questions raise on my side, that a further review of the mathematical derivation does not seem to to be possible any more. In the end, we have a one dimensional function in EQ 10, which is dependent on the constant parameters delta z, c, v*, which is extended with a variable delta z function for distances of more than 2D from the rotor. This function in EQ 14 should describe the radius of the core wake, which is untouched by the free-stream turbulence. However, no explanation how this function was derived is given.

While c may be computed more or less accurately from simulations and the sensitivity of v* on the result may be small as stated in lines 1-3, page 15, the parameter delta z should have a major influence on the modelled velocity deficit. Delta z is assumed to be the rotor radius, but no explanation is given for this. As delta z is defined as the height (measured from hub height), where the free-stream velocity is reached again, the rotor radius seems to be a choice, that does not comply with the reality.

Summarizing this part, considerable doubts on the physical assumptions, derivation and choice of parameters of the model must be raised. Dismantling these doubts would require a large effort and it is not entirely clear if this is possible. Therefore, I recommend to see the developed model as an empirical relation, rather then an analytical model. In this case, the derivation could be removed from the paper and the result could be stated without the claim of physical correctness.

Measurements

The description of the measurement setup and site as well as data acquisition seems a bit short to me. This means in particular:

- It is not clear what exactly represents u_r (see above). - It is not clear how u_r is

calculated from the measurements. The methodology how the velocity in the wake is calculated from the measurement signals should be explained at least briefly. In addition to that, the use of filters or similar of any kind should be mentioned. - It is not clear how u_0 is measured. Is there a met mast? Where is it? How long is the averaging time? What is the standard deviation? - Are there changes in u_0 during the experiment? - If u_0 is measured by a met mast (maybe at a larger distance), wouldn't is make sense to determine u_0 from the UAV measurements on the flight path in a certain distance to the wake? In this way, u_r/u_0 could always be computed with a continuously updated value. - The results of a comprehensive measurement campaign are reduced to some mean values. In order to judge on the quality of the measurements, the lateral velocity profiles should be included into the manuscript. This would also underline the scientific value of the measurements. - The operational state of the wind turbine is not mentioned. Is the turbine in below rated conditions? Were pitch angle and rotational speed constant for all measurement runs? - A discussion on the uncertainty of the measurements related to the actual measured velocities is missing. In a work by Subramanian the absolute uncertainty of the UAV wake velocity measurement is stated with 0.7m/s. Applying this to the measured wind speed at 1D in Figure 7, which is 0.3*u_0=3.15m/s, would yield an uncertainty of 22%. I recommend to insert a discussion on this. - From my understanding, the height of the flight paths should be more or less constant. What is the tolerance here? - It is explained, that the flight path during flight 1 is not suitable at some points, which leads to the exclusion of some measurement lines. However, there are also points missing, where the trajectory of path 1 seems to be very similar to the others (x=D and x =2D). Also other measurement points are from flight 2 and 3 are missing. It should be explained and at least exemplarily demonstrated why those measurement points were excluded.

General comments

- It is not clearly stated, what is the advantage of the developed analytical model in comparison to other models. However, it criticised that previously developed wake deficit

models do not take into account the atmospheric conditions. From my understanding, the present model includes this influence with the parameter u*. In the discussion, it is stated that the model is quite insensitive to this parameter. Does'nt this mean, that the present model is also not really including the influence of the ABL characteristics? - The literature review does not contain other measurement campaigns with UAVs. It is therefore quite difficult for a reader, who is not familiar with such kinds of measurements, to set the presented measurements into a context.

Conclusion

Concluding this review, a lot of minor and some major issues were identified. Some of the issues may be caused by misunderstandings, which in turn means that further explanations should be given. This is especially true for the derivation of the analytical model. From my point of view, the manuscript needs considerable reworking in order to gain a positive recommendation. However, if it is not possible to dismantle the doubts on the analytical derivation, the main original part of this work would be missing and another focus needs to be found.

Bartl et al., Wake measurements behind an array of two model wind turbines, Energy Procedia 24 ( 2012 ) 305 – 312, doi: 10.1016/j.egypro.2012.06.113

Kim et al., Wind Turbine Wake Characterization for Improvement of the Ainslie Eddy Viscosity Wake Model, Energies 2018, 11, 2823; doi:10.3390/en11102823

Parra et al., Momentum considerations on the New MEXICO experiment, The Science of Making Torque from Wind (TORQUE 2016), doi:10.1088/1742-6596/753/7/072001

Emeis, S., Wind Energy Meteorology, Springer, Heidelberg, Germany, 2017.

Subramanian et al., Drone-Based Experimental Investigation of Three-Dimensional Flow Structure of a Multi-Megawatt Wind Turbine in Complex Terrain, Journal of Solar Energy Engineering, OCTOBER 2015, Vol. 137 / 051007-1

---

## Author Comment (AC1) · 11 Sep 2020

**Author's response to Referee #1**

September 8, 2020

Thank you for the detailed review of the manuscript. In the following I will comment on each point. The referee's comments will be repeated in blue italic before the answer. We will adopt the enumeration format from the original referee's comment list.

**Summary and general comment**

*The authors present an equation to reproduce the wind deficit behind a wind turbine, and compare their results with unmanned aerial system (UAS) observations. I appreciate the general strategy of trying to establish a wake thickness description with a model based on momentum equation and checked against UAS observations. However I do not find the paper really convincing neither in the analytical modelling nor in the observational section. The observations presented in the paper are restricted to a few values of the mean wind measured by the UAS at several distances from the rotor. Furthermore, some key parameters, such as the friction velocity used by the authors in their parametrisation, are not computed from observations (or retrieved from simulations with a weather model), but arbitrarily fixed to a value of 0.3 m/s, supposed representative of a wide range of meteorological conditions. The mathematical developments are confusing, and, in my opinion, wrong in some parts. I find it hard to believe that all of the co-authors have carefully examined this manuscript before its submission to WES.*
*There is abundant literature on turbine wake observation and modelling (the authors mention the comprehensive review paper by Porté-Agel et al., which appeared this year in Boundary-Layer Meteorology). It is therefore crucial that any new article clearly explains what is brought with respect to the existing knowledge. To summarize, the present manuscript needs a lot of work, on both form and content, before becoming acceptable in the Journal.*
Thank you for sharing your general thoughts. Below we answer to all the raised issues.

**Major comments**

1. *I do not find any interest in the Euler method to solve the model equation. If there exist an analytical solution, why playing with approximate, numerical solving? This adds confusion.*
   In this manuscript the Euler method is intended as validation tool for the simplification done in Eq. 7 (or from Eq. 8.1 to 8.3). The differential equation (or to be more precise: difference equation) is a non-homogeneous DE. When integrating Eq. 8.1 we treat $u_r$ independent of $x$ which is not really correct. But it makes the solution much easier treating $u_r$ constant over (an infinitesimal increment) $dx$ . The quadratic character of the solution is kept, as can be seen in the resulting Eq. 8.3. Also, presenting a rather simple numerical method that solved the equation that can describe a wind turbine wake, can be motivation for an implementation in any numerical solver.

2. *A lot of analytical models describing the wind deficit behind a rotor are already available in the literature. The authors do not explain why there is a necessity for a new one, what is the improvement brought by their model, how it compares with the existing ones, etc.*
   Thank you. This is fair point that we will address in the new version of the manuscript.
   The authors of the manuscript are all affiliated with atmospheric (in-situ) measurements/science. In-situ measurements of the near wake are extremely rare in the scientific community and often overlooked (e.g. the comprehensive paper of Porté-Agel et al. (2020)). In the earliest evaluation period of the data, the in-situ measurements were plotted against models from Bastankhah and

Porté-Agel (2014, 2017) and Emeis (2010). Given that the Emeis model (E10) is intended for wind parks the idea was to modify the approach to fit a single wind turbine deficit. The analytical solution based on the model from Bastankhah and Porté-Agel (2014, 2017) did not fit to the measurements. The data could fit in the near wake or in the mid wake. But never over the whole measured wake region (0.5 - 5 D). We concluded that the simplifications made by Bastankhah and the origin (thrust based derivation) did not allow for a fit to real world data, e.g. neglecting tip vortex helix structures. Yet, we do not want to take anything away from thrust based models.
We will adjust the introduction to tell the reader which concerns lead to the alternative approach of a new analytical model.

3. *The hypotheses used in the equations lead to a mathematical impasse: it is assumed that the transversal and vertical wind components are zero ($\overline{v} = \overline{w} = 0$. We thus have $d\overline{v}/dy = d\overline{w}/dz = 0$, and to satisfy incompressibility in mass conservation equation, we therefore get $d\overline{u}/dx = 0$!. Furthermore, the authors come to the relation $\overline{u}d\overline{u}/dx = d\overline{u^2}/dx$ (equations (2) and (4)), in contradiction to the mathematical relation $d\overline{u^2}/dx = 2\overline{u}d\overline{u}/dx$.*

We have reworked this part of the manuscript. The main focus here shall be to condense the Navier-Stokes (N-S) equations to get to the momentum conservation equation in a steady-state incompressible flow (Eq. 1).
The presented N-S equation is already the divergence-free version. So we are very thankful to point out this error. We have now updated the steps to boil the N-S down to the momentum conservation equation for a steady-state incompressible flow.

The full derivation can be found in the new iteration of the manuscript. We believe that we have connected the N-S equation to the resulting Eqs. 8,11,12 in the manuscript successfully.

$$\underbrace{\frac{\partial(u_r \cdot u_r)}{\partial x}}_{A} + \underbrace{\frac{\partial(\overline{v} \cdot u_r)}{\partial y}}_{B} + \underbrace{\frac{\partial(\overline{w} \cdot u_r)}{\partial z}}_{C} + \underbrace{\frac{\partial(\overline{u'u'})}{\partial x}}_{D} + \underbrace{\frac{\partial(\overline{u'v'})}{\partial y}}_{E} + \underbrace{\frac{\partial(\overline{u'w'})}{\partial z}}_{F} = 0 \qquad (1)$$

4. *The authors mix partial derivative equations and bulk approximations with finite differences (e.g. Eq. 5). If an analytical solution is to be found, then the mathematical developments have to be conducted with the derivative forms (i.e. not approximate) of the equations.*
We know this is not hundred percent accurate. From our understanding the final analytical solution is derived from the governing NS equation. From thereon we have to do some simplification using real world order of magnitudes (e.g. $\Delta z$, or the Gradient approach in Eq. 5). We then stick to differences assuming a continuously differentiable solution/equation. We will mention this issue in the paragraph before continuing with Eq. 7.

5. *The manuscript reveals weaknesses in the knowledge of boundary-layer meteorology. It is mentioned that the model is applied above the surface layer ($d\overline{u}/dz = 0$ from the hub height), but the equation used to compute the eddy-diffusivity (Eq. 6) is valid in the surface layer (and for neutral conditions). A sentence such as "Regarding the temperature profile, wind conditions and turbulence intensity (s.a Sec. 3.3), a stability parameter $\zeta = z/L$ of approximately 0 to 1 can be concluded, using Businger et al. (1971)." is really annoying, because estimating the stability requires the knowledge of friction velocity and buoyancy flux, and none of these two parameters was measured during the observation periods.*
Equation 6 is indeed valid in the surface layer. From our understanding, up to the inversion height $z_i$. We (have to) assume the hub height of the wind turbine to be still in the surface layer, in order to be able to use this equation. As in Fig. 1 implied, we assume that $du/dz \approx 0$ not $= 0$. We are aware that we stretch the validity of some of the equation. It would have been really great to have had an EC (eddy covariance) station at the measurement site at the field campaign. But then, measurement campaigns are always more complex as people might assume. We have to work with what we got at the moment. And as mentioned in the discussion, if we would have known the outcome of the data evaluation, we would have installed an EC station or would have implemented a measurement path for the UAS to measure the horizontal Reynolds shear stress on site at a decent altitude. But wind parks (off- or on-shore) are never ideal to measure undisturbed flow. Hence, we use this method to at least somehow assume a $u_*$ value, for a near neutral thermal stratified lower

atmosphere, using the vertical profile of the virtual potential temperature directly in front of the wind turbine.

And please also consider that the mathematics or the validity of the derivation would not change by using a 'true' value for the shear velocity.

6. *The results presented in Figs. 7 to 10 should be grouped in a single graph (note that the observations presented in these 4 figures are the same). The curves relative to Euler's solutions should be omitted, as well as the model curves which are not relevant ($\alpha = constant$ for distances to the turbine larger than two diameters).*

Thank you. We can see, how the four figures really might be unnecessary. This is work flow related. As the manuscript evolved, the far wake behaviour of the model got interesting. The initial intention was to have graphs that depict the near wake behaviour and an overview ranging all the way up to the far wake of the wind turbine (although we have no UAS data). Yet, we wanted to see, if the model depicts a realistic wake length (what it does). But we admit that the near wake behaviour is still reasonably well shown in the far wake graphs. Concerning the suggestion to combine all graphs into one plot, we do not want to do so, since we think that the graphs will look cluttered, especially since we use the Euler method as validation for the analytical derivation of the model and need to keep it in the graphs. We oppose your suggestion to remove the Euler method completely (from the graphs and from the manuscript).

We also do not want to remove the $\alpha = constant$ line for distances larger than $2\,D$, because this is the essence of the graph. The result is that the $\alpha = constant$ line does not match the measured data. Removing this line would be contrary to the graph's intended statement.

**Specific comments and technical errors**

1. *There is no mention of the turbine parameters (e.g. the thrust coefficient), though some wake models involve such parameters in their equations. This should be commented.*

We have trouble to assign this comment to a specific occasion. We mention the models of Magnusson and Smedman (1999) and Bastankhah and Porté-Agel (2014) and that they use the thrust coefficient $C_T$. Maybe you mean that we do not mention turbine parameters in our model. We use the diameter of the wind turbine, but this is mentioned throughout the manuscript.

It is a unique selling point to contrast with existing models and should be more stressed in the introduction.

2. *P. 2, L. 19-21: Is there any specific interest to mention this study in regard to numerous other observations done in the wake of wind turbines?*

We aim for an overview of all methods that are used to measure wind turbine wakes and wind deficits. Hence, we want to mention that there are also approaches using remote sensing methods.

3. *P. 3, eq. (2): in the rhs, uj and ui should be overlined separately (i.e., with bracket notation, $\overline{u_j}\,\overline{u_i}$ instead of $\overline{u_j u_i}$).*

It is actually the case, but the 'overline' command is used instead of the 'bar' command. Each with their own problems. We have addressed the issue and added a spacer.

4. *P. 3, L. 10-12: If there is no pressure gradient, then there is no geostrophic wind, and in non-perturbed conditions the wind comes to zero. A geostrophic balance (compensation between pressure and Coriolis forces) should instead be invoked here.*

This might be a misunderstanding. We assume no *changes* in the pressure distribution along the wake. Meaning no change of the pressure gradient. We will make this clear in the next iteration of the manuscript.

5. *P. 3, lines 19-20: It should be explained why subsidence might be neglected in unstable and neutral conditions (implying a different behaviour in stable conditions?).*

Buoancy-driven flows need a variation in density (and a gravity field). In neutral conditions the BL can be assumed well mixed. In stable condition a WEC is mixing air over an altitude of up to 200 m, e.g. in modern off-shore wind parks. Here, the artificial and forced mixing of the air

can lead to subsidence. Especially, when the hub height of the wind turbine is somewhere near the inversion height $z_i$, e.g. during diurnal transition in a MBL.
We will add a short explanation.

6. *Fig. 1: The wind profile represents the conditions ahead of the wind turbine. A second profile representative of the wake conditions should be added.*
   This is a good idea. We have added the additional wind profile.

7. *To avoid confusion, I suggest to replace z (hub height) and $\Delta z$ with, e.g. h and $\Delta h$.*
   This is true. We can see the confusion. At the moment something like $z \to z + \Delta z$ is in place. We will implement a more comprehensive labelling of the heights in the new iteration of the manuscript.

8. *P. 4, L. 11: Typo "von Karman".*
   Fixed.

9. *P. 4, last line: "Joffre et al. (2001) studied the variability of the stable and unstable atmospheric boundary layer height and documented a dependence of the shear stress velocity $u_*$ on the stability parameter $\zeta$." The main driver of $u_*$ is the wind. Furthermore, $u_*$ is one of the two parameters used to define the stability (and not the other way).*
   Fair enough. We could change it up. However, this description is directly a citation of Fig. 2 in Joffre et al. (2001).

10. *10. P. 5, L. 2: "Slight differences in $u_*$ solely shift the solution along the y axis". Please explain. What is "the y-axis"?*
    The sentence has been removed. The sensitivity towards a change of the shear velocity $u_*$ will be addressed more comprehensively in the new manuscript.

11. *P. 5, L. 3: Unclear for me.*
    Please be more specific, so we can clear out the issue. We assume you might mean the need for the parametrization, so we added a short explanation.

12. *P. 5, L. 4: Typo "reduced".*
    Fixed. We have also moved the sentence up directly under Eq. 5 were $\Delta u$ appears for the first time.

13. *P. 5, eq. 7: Please define what $\alpha$ represents.*
    $\alpha$ is the wind deficit decay rate. We will add a mention.

14. *P. 5: "Equation 7 is a non-homogeneous non-linear differential equation (DE) of first order". As it stands, Eq. 7 is not a differential equation.*
    We changed the sentence to be more exact.

15. *P. 5, eq. 8.2: Please define what the superscript 'hom' represents.*
    It stands for 'homogeneous'. The issue has been cleared out.

16. *P. 5: The paragraph "A short assessment ... convenient to solve." Is unclear. Please rephrase.*
    The paragraph has been restructured and rephrased.

17. *P. 6, eq. 9 and 10: It is not useful to write two equations here.*
    We have omitted the former Eq. 10 and added some explanation.

18. *P. 6, L. 24: "the frequency $\alpha$". Why is $\alpha$ called a "frequency"?*
    Since it sounds a bit odd, we changed 'frequency' to 'wind deficit decay rate'.

19. *P. 6, eq. 11: Is there any justification (e.g. a reference) for this equation?*
    In this study we use this hyperbolic function as a first approach. Different functions (quadratic $(1/x^2)$ or any potential law function) were used, and the one in the submitted manuscript worked best. The justification behind it, is the idea that the function should somewhat represent a non-linear decay.

20. *P. 6, eq. 11: There is no need to introduce a new symbol (R), since D=2R. Please rewrite as a function of D.*
    The equation has been adjusted.

21. *P. 7, L. 10: typo "... the this method".*
    Thanks. It has been fixed.

22. *Fig. 3: Please define clearly which parameter is represented here.*
    The caption of Fig. 3 has been rephrased.

23. *P. 10, L. 3: UAS should be defined at its first appearance (p. 2).*
    You are correct. No argument here.

24. *P. 10, L. 5: What are the heights of the legs?*
    All data were captured at hub height. The information has been added. Additionally the information is now also given in the caption of Fig. 5.

25. *P. 10, L. 11: Please explain how the turbulence intensity is computed (turbulence observations are not mentioned in the manuscript).*
    This information is from the vertical profile flown in the inflow of the WEC. It is computed by calculating $TI = u'_{hor}/\overline{u_{hor}}$. With $u_{hor}$ the horizontal wind. The averaging window is derived from the computed integral length scale. The information has been added to the manuscript.

26. *P. 10, L. 19: Typo "parameters".*
    Thank you. Fixed.

27. *Fig. 5: Please add a scale and indicate the geographical orientation. We can observe close to the right border of this image the shadow of a second wind converter. Is there a potential impact of this 2nd converter on the wake of the 1st one?*
    We will add a new map for Fig. 5. The WEC south of the E-112 converter did not interfere with the measurements. Due to the wind conditions the UAS needed extra space for a wider turning manoeuvre.

28. *There are negative altitudes. Please explain.*
    The altitude is actually m a.s.l not a.g.l, and the issue has been addressed.

29. *Fig. 6: Please explain why temperature data are discarded during UAS turns. Is that because the measurements are biased, or because turns are too far away from the profile location?*
    Thank you, that is an interesting topic in itself. The data needs to be discarded, because of the internal boundary layer that touches the temperature sensor. Thus, the flow conditions are no longer according to the specifications of the temperature measurements. The flow temperature is then highly impacted by the temperature of the fuselage of the UAS.
    We have also added a short explanation in the caption of Fig. 6.

30. *Fig. 7, caption: replace "At about 2.5 D..." with "From about 2.5 D..."*
    Fixed.

31. *Section 4.3: There are no observations in the far wake area. So, the model performance cannot be evaluated. Why do not try to test the model against another data set?*
    As you might have noticed there is a far wake model behaviour analysis later in the manuscript. At the measurement campaign we had a tight schedule, since we have to perform several measurement flights for different wake evaluations, amongst them also measurements for the partners of the HeliOW project.
    We are planning to measure at a similar data set behind a E-82 WEC in the south of Germany. We have also applied the model to wake data from off-shore wind parks, where it performs very well.

32. *Section 4.3: The sentence "While the constant-$\alpha$ model underestimates the wake behaviour the dynamic-$\alpha$ approach follows the measured data up to 5 D and paints a reasonable picture of the wind deficit decay." is not relevant for this section.*
    You are correct. We will move the evaluating part of the statement to the discussion section.

33. *Figs. 7 to 10: The parameter represented is not the "wind deficit".*
    Strictly spoken it is the normalised reduced (or residual) horizontal wind speed in the wake. We choose the remaining wind velocity to represent the wind deficit in the wake. To prevent confusion we will call it as such. It is much more convenient to work and calculate with what is left than what is missing in wind velocity.

34. *P. 14, L. 3-5: I do not understand what is meant here. Please rephrase.*
    Fair point. We have stretched the paragraph and explained the matter more thoroughly.

35. *P. 15, L. 4: "0.45 m/s".*
    Thanks. It has been fixed. The paragraph has been altered altogether.

36. *P. 15, L. 1 to 5: This is surprising: it is known that the greater the turbulence level, the shorter the wind recovery distance in the wake. Furthermore, if u\* is a key parameter in the eddy-diffusivity value, then enhancing or lowering it by 50% should significantly modify the wake characteristics.*
    True. The paragraph is outdated. It refers to a first evaluation that only considered up to 5 D, and no wake lengths at all. In an up-to-date evaluation it could be shown that the wake length differs $\pm 3$ D for the assumed variations in $u_*$ of $\pm 50\%$. Thus, this paragraph has been rewritten.

37. *Fig. 10: The curve corresponding to the analytical model here is not identical to that presented in Fig. 8. For example, at a distance of a little less than 5D, the model crosses the blue disk of the observations in Fig. 10, whereas it remains well below in Fig. 8. Please explain.*
    We have addressed the issue. The wrong plot/pdf was copied into the tex folder. The plot was from an early calculation where the reference height ($z = h + \Delta z$) was wrongly implemented ($\Delta z$ was missing).

**References**

Bastankhah, M. and Porté-Agel, F. A new analytical model for wind-turbine wakes. *Renewable Energy*, 70:116 – 123, 2014. ISSN 0960-1481. doi: https://doi.org/10.1016/j.renene.2014.01.002. Special issue on aerodynamics of offshore wind energy systems and wakes.

Bastankhah, M. and Porté-Agel, F. A new miniature wind turbine for wind tunnel experiments. part ii: Wake structure and flow dynamics. *Energies*, 10:923, 07 2017. doi: 10.3390/en10070923.

Emeis, S. A simple analytical wind park model considering atmospheric stability. *Wind Energy*, 13(5): 459–469, 2010. doi: 10.1002/we.367. URL https://onlinelibrary.wiley.com/doi/abs/10.1002/we.367.

Joffre, S., Kangas, M., Heikinheimo, M., and Kitaigorodskii, S. Variability of the stable and unstable atmospheric boundary-layer height and its scales over a boreal forest. *Boundary-Layer Meteorology*, 99:429–450, 06 2001. doi: 10.1023/A:1018956525605.

Magnusson, M. and Smedman, A.-S. Air flow behind wind turbines. *Journal of Wind Engineering and Industrial Aerodynamics*, 80(1):169 – 189, 1999. ISSN 0167-6105. doi: https://doi.org/10.1016/S0167-6105(98)00126-3.

Porté-Agel, F., Bastankhah, M., and Shamsoddin, S. Wind-turbine and wind-farm flows: A review. *Boundary-Layer Meteorology*, 174:1–59, 2020. doi: https://doi.org/10.1007/s10546-019-00473-0.

---

## Author Comment (AC2) · 11 Sep 2020

**Author's response to Referee #2**

September 8, 2020

Thank you for the effort that has been put into this detailed review of the manuscript. In the following I will comment on each point. The referee's comments will be repeated in blue italic before the answer. We will adopt the enumeration format from the original referee's comment list.

**Summary**

*In the manuscript, the derivation and validation of a model for the velocity deficit in the wake of a wind turbine is presented. The model derivation starts from the Reynolds decomposition of the differential momentum equilibrium in a fluid and models a momentum flux from the wind at greater heights, which finally compensates the wake velocity deficit at a certain stream-wise distance to the wind turbine. A differential equation is obtained from the derivation and is solved analytically as well as numerically, where the analytical solution could only be obtained by introducing a simplification. Measure- ments of the mean wind speed in the wake using an UAV were undertaken to provide validation data to the derived velocity deficit model. The UAV was equipped with a five-hole probe for the velocity measurement. A flight pattern with 8 horizontal lines parallel to the rotor plane in different distances up to 5D from the rotor was chosen and repeated 3 times. The analytical as well as the numerical solutions of the derived differential equation was compared to the (mean) wake velocities obtained from the measurements. Good agreement was stated up to a distance of 2-3D behind the rotor. After this, the authors claim that the helical tip vortex structure has collapsed and therefore a modification of the derived velocity deficit model is presented. This modification is based on the assumption that a stronger mixing of the wake and the surrounding wind field is apparent from this distance. The modification of the model yields results that better fit the experimental data at higher distances. A discussion on the influence of the shear velocity, which is used as an input parameter of the velocity deficit, is added. In the conclusion, it is stated that the modelled and measured velocity deficit in the wake fit well and a number of possible improvements and further applications of the model are listed.*

**Comments**

1. *Before starting with the detailed comments, one major issue needs to be addressed:*
   *The variable $u_r$ is defined as "the reduced horizontal wind speed in the wake along the x direction". This definition is not sufficient. I assume that $u_r$ is the mean value of the wake velocity at hub height. All my comments are based on this assumption. Furthermore, it is not clear if the averaging length is one rotor diameter or if the wake expansion is considered (resulting in an increasing length of the averaging space with higher distances from the rotor). Applying the above assumed definition of $u_r$, the analytical model in Figure 7 shows a reduction of the wind speed in the wake of 70% at 1 D behind the rotor. This is within the scatter of the measurements. This seems to me a surprisingly low mean axial velocity in the wake for a normal operation of the rotor. In wind tunnel measurements of Bartl et al. we see a deficit of 40-50% at that point. Other wind tunnel measurements of Kim et al. show a similar picture at $\approx 1.5$ D with a deficit of a bit more then 40%, while the derived model shows a deficit of more than 60%. PIV measurements performed during the MEXICO experiment also show a considerably lower velocity deficit at design TSR (see Parra et al.). Especially when considering that a higher velocity deficit would be expected due to the absence of atmospheric turbulence in the wind tunnel experiments, the observed and calculated*

*velocity deficit in this work seem surprising to me. From my point of view, it needs to be clarified if this is really the case or if there is a misunderstanding on my side. If not, a discussion on this discrepancy is necessary.*

Your assumption is correct. In addition, we have defined $u_r$ more precisely in the next iteration of the manuscript.

Concerning the wind deficit measurements. The study by Bartl et al. (2012) states the residual wind speed in the wake (around 40%), as do our measurements; (we are aware of some confusing caption in Fig. 7 to Fig. 10 stating 'wind deficit measurements', as it is indeed the residual wind speed. We have addressed the issue). Also the wind tunnel experiments by Medici and Alfredsson (2006) show residual wind speeds between 20-30%. We think that Kim et al. (2018) average over the whole wake length behind the nacelle. This is also not representative of the wind deficit, since the undisturbed flow can stream around the nacelle and increases the residual wind speed behind it (nacelle effect). In our study, we want to consider the part of the flow (and its velocities) that represent the actual energy conversion. Wildmann et al. (2014) uses in-situ UAS measurements and shows a wind deficit of about 60% behind a Kenersys 2.4 MW converter.

Canadillas et al. (2020); Siedersleben et al. (2018) introduced a minimum average method for wind deficit calculation behind wind parks where the wake may meander. While this is not the case for our measurements, the same idea is implemented. We average around the wind speed minimum in the wake (neglecting blade-tip influence). Blade-tip vortices superimpose (positive and negative) with the horizontal wind speed at hub height. Thus they can have an influence on the measurement of the horizontal wind Mauz et al. (2019) at the borders of the wake to the undisturbed flow.

While this matter seems to be controversial, we want to argue that whatever averaging method might be applied, the model could adopt to the change of the wind deficit (being it 40% or 60%). We believe we have chosen an average method representing the energy conversion from the free flow, the most. The main argumentation in our manuscript, the change of atmospheric inflow along $x$, being caused by the collapse of the tip-vortex helix at around 2-3 D, remains unaffected. Regardless of the averaging technique.

2. *The general idea of the manuscript and the measurements seem promising to me, but the implementation and description of the performed work lacks accuracy at some points, which makes it difficult to judge on the results*

   Thank you. While this comment is very vague, we will try to be more accurate in the next iteration of the manuscript. We have added some more paragraphs where we thought a reduction in the proceeding speed would be beneficial to the reader.

3. *The comments will be clustered in three groups, namely: Derivation of the analytical velocity deficit model, Measurements, General comments.*

   We will adopt the commenting format of the referee in this response. To enhance the overview over all comments and to ease future reference, in addition we have enumerated the comments.

**Derivation of the analytical velocity deficit model**

1. *The derivation starts promising with a description of the Reynolds decomposition of the differential momentum equilibrium in a fluid. However, the equation is dramatically reduced by a number of assumptions. After this, the remaining ($u'w'$) term is replaced by an empirical relation. Here, the derivation starts to become difficult to understand and seems to contain some mathematical mistakes or some steps of the derivation were skipped, which prevents the reader from understanding what exactly happened here.*

   We have restructured this part of the manuscript in the new iteration and added some more descriptions. From Eq. 7 to Eq. 8.3 we have a very detailed explanation of each step. We also mention the simplification we made, and how we justify to do so. Beyond the justification we also show, by using the numerical solution (Euler method), that the simplification can be done.

   Later on, solving the quadratic solution is not presented in any more forms or details, since we believe it to be trivial. Unfortunately, we can only speculate what to improve, since you do not specify what seems to be the mathematical error.

2. *The reduction of the momentum equation is based on several assumptions. The assumption of a 'one dimensional, horizontal steady-state wind field' implies that the wind turbine wake is no longer seen as a three dimensional tube or something similar. The model therefore assumes that a momentum flux can only be added to the wake region from higher altitudes but not from the flow on the left and the right from the wake. This assumption is valid for the far wake of wind farms, where the velocity deficits of multiple wind turbines merge and a more or less homogeneous horizontal layer with a velocity deficit up to a small height (in the order of magnitude of the wind turbine height) in comparison to its lateral size (in the order of magnitude of the wind park width) can be assumed. Here, the influence of the added momentum from the sides is negligible. This is not the case for a single wind turbine and no explanation why this assumption should be valid was found. In addition to that, the authors apply this assumption to the near and mid wake region, which is a region, where the flow is strongly dominated by the geometry of the tip vortex structure. These vortex structures seem completely neglected in this approach.*

Thank you. We have also considered a radial symmetric approach. Then, $\Delta z \to r$, the radius of the rotor plane. The problem that arises is that we then have to specify the lateral momentum flux. While we think in reality there is one. This flux is caused by the WEC itself, while the downward facing momentum flux is a (more or less) constant atmospheric parameter (Regardless of the presence of the WEC). So we want to argue that the wind deficit decay may be influenced in part by lateral momentum flux, but the momentum sink aloft the WEC is the main driver of the wind deficit decay. We also only consider the centre line of the wake, assuming the wind energy conversion is taking place along the centre line (which it does not exactly).

Let us consider a radial symmetric ($\approx$ rectangular, for the sake of flux calculations) approach, then Eq. 4 in the manuscript would be:

$$\frac{\partial u_r^2}{\partial x} + \frac{\partial (\overline{u'w'})}{\partial z} + 2\frac{\partial (\overline{u'v'})}{\partial y} = \frac{\partial u_r^2}{\partial x} + \frac{\partial (\overline{u'w'})}{\partial r} + 2\frac{\partial (\overline{u'v'})}{\partial r} = 0 \tag{1}$$

$\partial r$ can then be substituted with $\partial z$. In our argumentation $\overline{u'v'}$ is neglectable. However, we incentivise a comprehensive field study to measure these fluxes next to a WEC and also in the undisturbed flow. We think this is one of the main goals for a scientific study and a future improvement of the model. If $\overline{u'v'}$ and $\overline{u'w'}$ would be measured precisely and reliably, one could even think about solving the above equations for each $x/D$, without substituting the Reynolds stress term(s). But this is something we learnt after evaluating all data. For now, we have to deal with the method presented in this manuscript.

As you stated in your first comment, we replace $\overline{u'w'}$ by an empirical relation (Gradient method). Here, we stretch the validity of Eq. 6 ($K_m = \kappa \cdot u_* \cdot z$), which we will mention in the new manuscript. However, what it then comes down to is to find a value for $u_*$. Ideally, a method to calculate $K_m$ at hub height would be great. But the measurements were not really suited to do so (e.g. obstacles on the ground like dike, high-voltage power lines, vegetation and industrial buildings in the inflow, resulting in no available free ranging flight path in the undisturbed flow). Also, we could not find a method to calculate $K_m$ at hub height. $u_*$ is defined for a surface measurement, and a multiplication by $z$ states a linear increase with height (in the Prandtl layer). So, since we stretch the applicability of this equation by multiplying with a value for $z$ (the hub height $h$ what implies that we assume the WEC to be still in the surface layer), we receive an over estimate of $K_m$. This means in return, that we under estimate $\Delta z$, once we fit the model to the data ($\alpha = K_m/\Delta z^2$). To battle this dilemma we made the argument that $\Delta z$ shall be the rotor radius of the WEC. From thereon the only parameter to influence $\alpha$ is $K_m$. Its determination then is described in the manuscript using the vertical virtual potential temperature profile, to derive a reasonable and typical $u_*$ for these atmospheric conditions. It is easy to argue, but for now very hard to proof that $u_*$ should be smaller and $\Delta z$ be larger. In the end, it would not effect the value of $\alpha$. This is were the measurements come in and show their value as the foundation for the model (fitting). They allow the determination of $\alpha$ using the assumptions stated above. Alternatively, one could even simply chose any numeric value as $\alpha$ and use any best fitting method. Yet, we believe, we have done the best, to back the calculation of $\alpha$ scientifically.

On a side note:

The determination of $\Delta z$ in this model approach, but also in the Emeis (2010, 2017) (E10) model, is one of the remaining scientific tasks (s.a. Platis et al. (2018)) which once solved, will improve the model and all the statements that can be derived by its results (e.g. internal boundary layer heights above wind parks, influence of inversion height in wind wakes etc.).

3. *After reducing the momentum equations, the term $(u'w')$ in Eq. 4 is replaced by an empirical correlation, which is inspired by the work of Emeis. $(u'w')$ is set to a term stated by Emeis that models the momentum flux from the above air layers into the wake. In Emeis work, this term is used to compute the integral (from free-stream to hub height) momentum flux. However, Eq. 4 is derived from the momentum equilibrium in its differential form, meaning that no integration over the height took place. It is not clear, why this should be valid. This problem is also visible, when differentiating $(u'w')$ by the z coordinate in Eq. 7. From my understanding of the derivation, this is simply done by dividing the equation by $\Delta z$. $\Delta z$ is defined as the vertical distance of the hub to a flow layer, where no velocity deficit is present. I could not figure out, how the differentiation of the expression in Eq. 5 representing the integral momentum flux over the height can lead to this expression. Furthermore, it seems that Eq. 7 shows a difference quotient instead of a derivative, which requires a solid explanation. In addition to that, the function shown in Eq. 7 seems to be independent from the height, as $\Delta z$ is a constant as described in line 5, page 4., while Eq. 4 is not defined for a certain height. It therefore needs to be clarified if Eq. 7 should be an evaluation of Eq. 4 at a certain height (including an explanation why this is done).*

We cannot find any integration in Emeis (2017). Emeis (2017) uses essentially the same approach up to Eq. 7 where he substitutes $dx = dt/du$, with $du = u_0 - u_r$.

Concerning the $\Delta z$ confusion. We do not divide any Eq. by $\Delta z$. Eq. 4 is taken, then we use a first order approach and assume continuous differentiability (which we will mention in the next iteration and try to make it more clear) and go from differentials to differences (bulk parametrisation). Now, we insert Eq. 5 into Eq. 4 and get Eq. 7.

As also mentioned above on a previous bullet point, $K_m$ is a function of height $z$. You are correct that Eq. 7 is then only valid at hub height. We shall mention this in the new manuscript.

4. *In Eq. 8.1 an integration is performed after rearranging the $\Delta x$ to the right side. Here, it is still not clear if $(\Delta u_r/\Delta x)$ is a derivative or a difference quotient. It is stated that both sides of the equation will be integrated, but the integration variable is not known. Assuming that $x$ is the variable to integrate over, the dependence of $u_r$ in the denominator of the first term in the braces seems to be ignored.*

$\Delta u_r/\Delta x$ is the result after going from $\partial$ to $\Delta$. We have changed it up in the new manuscript and do now use $d$ instead. We are dealing with first order approaches here (Gradient method in Eq. 5). We will state this more clearly, and then it should be fine.

Concerning ignoring the dependency of $u_r$ along $x$. We have to disagree. In the next 10 lines following the said integration we explain thoroughly why we simplify this integration and how we deal with it. We introduce a numerical validation calculation (Euler method) to estimate the error that we introduce by treating $u_r$ as a constant over $\Delta x$. This is the Eulerian analogy of the Lagrangian simplification done by Emeis (2017) considering the air parcel travelling at constant velocity through the wake, when solve the time dependent exponential solution behind the wake (e.g. for calculating velocities at distant $x/D$ in the wake of a wind farm as in Platis et al. (2018); Siedersleben et al. (2018)).

5. *At that point, so many questions raise on my side, that a further review of the mathematical derivation does not seem to to be possible any more. In the end, we have a one dimensional function in Eq. 10, which is dependent on the constant parameters $\Delta z$, $c$, $u_*$, which is extended with a variable $\Delta z$ function for distances of more than 2D from the rotor. This function in Eq. 14 should describe the radius of the core wake, which is untouched by the free-stream turbulence. However, no explanation how this function was derived is given.*

Okay. We can see that Eq. 14 (or Eq. 11) need a bit more of an introduction in the manuscript. The motivation for Eq. 11 is, that we needed a function that somehow represents the decay of the remaining WEC turbulence along $x$. In a first approach, we used a hyperbolic function to implement an asymptotic method for the remaining turbulence. Investigating the change of the decay

rate along $x$ is a goal for a stand-alone study, facilitating wake measurements along the whole wake (e.g. 1 - 10 D). We have added a brief explanation alongside Eq. 11.

6. *While c may be computed more or less accurately from simulations and the sensitivity of $u_*$ on the result may be small as stated in lines 1-3, page 15, the parameter $\Delta z$ should have a major influence on the modelled velocity deficit. $\Delta z$ is assumed to be the rotor radius, but no explanation is given for this. As $\Delta z$ is defined as the height (measured from hub height), where the free-stream velocity is reached again, the rotor radius seems to be a choice, that does not comply with the reality.*

Thank you for the comment. The parameter c does not need a computation. It can be measured or a literature value (e.g. Betz' law) can be used. This is indeed not trivial. Choosing $\Delta z$ or even to define where it should begin or end is not easy to determine (s.a. Platis et al. (2018)). Please keep in mind that we need to simplify reality. So, we assume an instant velocity jump at the border between the wake and the undisturbed flow. Consequently, in the model, the momentum in-flux comes directly from the layer aloft the WEC wake. In reality, there might be a multilayered internal boundary layer (especially at wind parks) aloft the wake. We agree that the parameter should be more discussed.
We will add an explanation in the manuscript. We also discussed the matter of $\Delta z$ in a previous comment.
The sensitivity of the model toward $u_*$ is completely reworked in the new manuscript.

7. *Summarizing this part, considerable doubts on the physical assumptions, derivation and choice of parameters of the model must be raised. Dismantling these doubts would require a large effort and it is not entirely clear if this is possible. Therefore, I recommend to see the developed model as an empirical relation, rather then an analytical model. In this case, the derivation could be removed from the paper and the result could be stated without the claim of physical correctness.*

Since this point is a summary of the previous bullet points we do not see an explicit need to address the issues, once again. Yet, we want to respond. A lot of the issues pointed out by referee #2 are legit and need to be addressed. However, some of which are based on misunderstandings. We have put a lot of effort to lay a physical foundation for the model. The mathematical simplifications are validated by the Euler method. We have used all scientific tools available to us (physical, mathematical and numerical instruments fit together). The in-situ measurements can be seen as proof for the physical correctness. However, we can see that the fact that the measurements do not cover the whole wake length may raise some concern.

**Measurements**

*Note: For the sake of clarity the authors restructured the measurement comments section to answer the raised issues more or less individually.*

*The description of the measurement setup and site as well as data acquisition seems a bit short to me. This means in particular:*

1. *It is not clear what exactly represents $u_r$ (see above). It is not clear how $u_r$ is calculated from the measurements. The methodology how the velocity in the wake is calculated from the measurement signals should be explained at least briefly. In addition to that, the use of filters or similar of any kind should be mentioned.*

We want to refer to Comment #1 above, where we have already explained the circumstance. The manuscript has also been updated to clear the issue for future readers.

2. *It is not clear how $u_0$ is measured. Is there a met mast? Where is it? How long is the averaging time? What is the standard deviation?*
*Are there changes in $u_0$ during the experiment?*
*If $u_0$ is measured by a met mast (maybe at a larger distance), wouldn't it make sense to determine*

[Figure]

Figure 1: Horizontal wind velocity ($v_h$) behind the E-112 converter at $x/D = 2$. The UAS data is smoothed out with a moving average over 50 data points to reduce turbulence effects and make an evaluation easier. The averaged reduced wind speed is calculated by $u_r/u_0$ (red and blue line).

*$u_0$ from the UAV measurements on the flight path in a certain distance to the wake? In this way, $u_r/u_0$ could always be computed with a continuously updated value.*

Sadly there is no met mast in the area available. But it also is not necessary. We use the UAS data to compute a value for $u_0$. Therefore, as you already suggested, an updated value is calculated for each flight leg (meaning for each distance $x/D$). This is beneficial, since the mean horizontal wind always varies a bit. In this study the horizontal wind deviated around $\pm 1.5 - 2$ m s$^{-1}$ (UAS data). The undisturbed mean horizontal wind has to be calculated with what is left of the flight leg and reaches into the free flow at hub height. This can be 100 m or 20 m. The analytical model is set up with an average horizontal wind $u_0 = 10.5$ m s$^{-1}$.

In addition we got the SCADA data (10 min averages) from the manufacturer. This data needs to be treated highly classified and the manufacturer also does not want to be mentioned in the paper. The average wind speed on top of the nacelle measured with a sonic anemometer is $\approx 10$ m s$^{-1}$. However, this measurement is biased by an internal boundary layer around the nacelle and can only be used as a rough estimate. The maximum value measured on top of the nacelle has been 12.7 m s$^{-1}$. Yet, the wind velocity variations comply with the UAS measurements.

3. *The results of a comprehensive measurement campaign are reduced to some mean values. In order to judge on the quality of the measurements, the lateral velocity profiles should be included into the manuscript. This would also underline the scientific value of the measurements.*

We have a plot for each flight leg. Here, an exemplary wake measurement evaluation plot is added (s.a. Fig. 1). When adding all plots to the manuscript, clarity will suffer, we suspect. However, we have implemented a data evaluation description in the manuscript.

4. *The operational state of the wind turbine is not mentioned. Is the turbine in below rated conditions? Were pitch angle and rotational speed constant for all measurement runs?*

The measurement flights 1-3 needed about 20 min. In this time span the wind turbine did not change its conditions. The E-112 WEC was operating near its rated conditions (rated wind speed: 13 m s$^{-1}$) with an average horizontal wind speed of 10.5 m s$^{-1}$. The average blade angle from SCADA is 1° with a rotational speed of $\approx 11.7$ rmp.

5. *A discussion on the uncertainty of the measurements related to the actual measured velocities is missing. In a work by Subramanian the absolute uncertainty of the UAV wake velocity measurement is stated with* $0.7$ m s$^{-1}$. *Applying this to the measured wind speed at 1D in Figure 7, which is* $0.3 * u_0 = 3.15$ m s$^{-1}$, *would yield an uncertainty of 22%. I recommend to insert a discussion on this.*

The manufacturer of the five-hole probe claims an accuracy of $0.1$ m s$^{-1}$ Rautenberg et al. (2019). In-flight conditions may vary this value a bit. But in general with the improvements made in design, IMU and GPS positioning an in-flight accuracy of $0.2$ m s$^{-1}$ can be expected. The components used in MASC-3 and the aircraft design can not be compared to the the UAV by Subramanian et al. (2015). Also path accuracy is an important parameter when calculating the 3D wind vector. See also next comment. We have added a short error consideration in the discussion.

6. *From my understanding, the height of the flight paths should be more or less constant. What is the tolerance here?*

Yes, the flight path is more or less constant and tracked by the auto-pilot. An accuracy of $\pm 2.5$ m in altitude deviation is achieved. This also depends on the level of turbulence. The movement of the UAS (e.g. up-down acceleration) is logged and also accounted for in the 3-D wind measurement calculations. This information has been added to the manuscript.

7. *It is explained, that the flight path during flight 1 is not suitable at some points, which leads to the exclusion of some measurement lines. However, there are also points missing, where the trajectory of path 1 seems to be very similar to the others ($x = D$ and $x = 2D$). Also other measurement points are from flight 2 and 3 are missing. It should be explained and at least exemplarily demonstrated why those measurement points were excluded.*

At $x = 2$ D there is too much variance in the velocity measurement, therefore a clear statement can not be made. We suspect tip vortex influence while entering and leaving the wake together with a too short flight path prohibit a reasonable determination of $u_r/u_0$.
Regarding the missing point at $x/D = 1$, we are very thankful. There was a decimal error in the used data frame. Because we calculated the residual wind velocity to $0.4 u_0$ at 117 m behind the WEC (it was set to $4 u_r$).
The data have been checked for similar errors. Non were found. We will update the graphics.
Concerning flights 2 and 3: the flight track alone can not be seen as an indicator for a successful measurement. At the day of the measurement, the strong wind made manoeuvring in and out of the wake in a confined region more difficult as anticipated (hence the flight path adjustment). For a successful measurement the yaw, pitch and roll angle also must be up to specs. So it can happen that due to a tip vortex hit the calibrated angles (and pressures) for the five-hole probe are out of their specifications. This can lead to NAs in the measurement. It is rare, but it happens.
Other measurements could have been used, if the flight path would have been longer. The flight pattern was set up for a wind direction of 90°. But from planning the flight pattern to starting the UAS and the measurement the wind direction changed slightly. Therefore, some measurements where corrupted by a too early turn of the UAS.

**General comments**

1. *It is not clearly stated, what is the advantage of the developed analytical model in comparison to other models. However, it criticised that previously developed wake deficit models do not take into account the atmospheric conditions. From my understanding, the present model includes this influence with the parameter u\*. In the discussion, it is stated that the model is quite insensitive to this parameter. Doesn't this mean, that the present model is also not really including the influence of the ABL characteristics?*

Thank you. A similar issue was raised by referee #1. The introduction has been complemented with a clear motivation for this model. We also have added a complete reworked sensitivity study concerning $u_*$. The previous discussion around a change in $u_*$ was considering distances up to 5 D where the differences are not that significant. When including the far wake, there are considerably

differences to see in wake length. This is also the expected behaviour. In the next iteration of the manuscript the new related section is redone.

2. *The literature review does not contain other measurement campaigns with UAVs. It is therefore quite difficult for a reader, who is not familiar with such kinds of measurements, to set the presented measurements into a context.*

   We have added information/literature of previous UAS measurement campaigns (e.g. in the introduction). We simply do not like to seemingly bloat the reference list.

**Conclusions**

*Concluding this review, a lot of minor and some major issues were identified. Some of the issues may be caused by misunderstandings, which in turn means that further explanations should be given. This is especially true for the derivation of the analytical model. From my point of view, the manuscript needs considerable reworking in order to gain a positive recommendation. However, if it is not possible to dismantle the doubts on the analytical derivation, the main original part of this work would be missing and another focus needs to be found.*

At this point we want to thank you for the detailed review again. It is apparent that you have put a lot of effort into following the manuscript in its first iteration. Unfortunately, the manuscript was not as easy to follow as we have wished. But we think with the points raised in this review – and the answers as well – the manuscript gained a lot.

**References**

Bartl, J., Pierella, F., and Sætrana, L. Wake measurements behind an array of two model wind turbines. *Energy Procedia*, 24:305 – 312, 2012. ISSN 1876-6102. doi: 10.1016/j.egypro.2012.06.113. URL http://www.sciencedirect.com/science/article/pii/S1876610212011538. Selected papers from Deep Sea Offshore Wind R&D Conference, Trondheim, Norway, 19-20 January 2012.

Canadillas, B., Foreman, R., Barth, V., Siedersleben, S., Lampert, A., Platis, A., Djath, B., Schulz-Stellenfleth, J., Bange, J., Emeis, S., and Neumann, T. Offshore wind farm wake recovery: Airborne measurements and its representation in engineering models. *Wind Energy*, 23, 02 2020. doi: 10.1002/we.2484.

Emeis, S. A simple analytical wind park model considering atmospheric stability. *Wind Energy*, 13(5):459–469, 2010. doi: 10.1002/we.367. URL https://onlinelibrary.wiley.com/doi/abs/10.1002/we.367.

Emeis, S. *Wind Energy Meteorology*. Springer, Heidelberg, Germany, 2017.

Kim, H., Kim, K., Bottasso, C., Campagnolo, F., and Paek, I. Wind turbine wake characterization for improvement of the ainslie eddy viscosity wake model. *Energies*, 11:2823, 10 2018. doi: 10.3390/en11102823.

Mauz, M., Rautenberg, A., Platis, A., Cormier, M., and Bange, J. First identification and quantification of detached-tip vortices behind a wind energy converter using fixed-wing unmanned aircraft system. *Wind Energy Science*, 4(3):451–463, 2019. doi: 10.5194/wes-4-451-2019. URL https://www.wind-energ-sci.net/4/451/2019/.

Medici, D. and Alfredsson, P. H. Measurements on a wind turbine wake: 3d effects and bluff body vortex shedding. *Wind Energy*, 9(3):219–236, 2006. doi: 10.1002/we.156. URL https://onlinelibrary.wiley.com/doi/abs/10.1002/we.156.

Platis, A., Siedersleben, S. K., Bange, J., Lampert, A., Bärfuss, K., Hankers, R., Canadillas, B., Foreman, R., Schulz-Stellenfleth, J., Djath, B., Neumann, T., and Emeis, S. First in situ evidence of wakes in the far field behind offshore wind farms. *Scientific Reports*, 8:2163, 2018. doi: 10.1038/s41598-018-20389-y.

Rautenberg, A., Schön, M., zum Berge, K., Mauz, M., Manz, P., Platis, A., van Kesteren, B., Suomi, I., Kral, S. T., and Bange, J. The multi-purpose airborne sensor carrier masc-3 for wind and turbulence measurements in the atmospheric boundary layer. *Sensors*, 19(10), 2019. ISSN 1424-8220. doi: 10.3390/s19102292. URL `http://www.mdpi.com/1424-8220/19/10/2292`.

Siedersleben, S. K., Platis, A., Lundquist, J. K., Lampert, A., Bärfuss, K., Cañadillas, B., Djath, B., Schulz-Stellenfleth, J., Bange, J., Neumann, T., and Emeis, S. Evaluation of a wind farm parametrization for mesoscale atmospheric flow models with aircraft measurements. *Meteorologische Zeitschrift*, 27(5):401–415, 12 2018. doi: 10.1127/metz/2018/0900.

Subramanian, B., Chokani, N., and S. Abhari, R. Drone-based experimental investigation of three-dimensional flow structure of a multi-megawatt wind turbine in complex terrain. *Journal of Solar Energy Engineering*, 137:1007–1017, 07 2015.

Wildmann, N., Hofsäß, M., Weimer, F., Joos, A., and Bange, J. MASC; a small Remotely Piloted Aircraft (RPA) for wind energy research. *Advances in Science and Research*, 11:55–61, 2014. doi: 10.5194/asr-11-55-2014. URL `http://www.adv-sci-res.net/11/55/2014/`.